# Asymptotically Fast Clebsch-Gordan Tensor Products with Vector Spherical Harmonics

**YuQing Xie** [*][1]  **Ameya Daigavane** [1]  **Mit Kotak** [2]  **Tess Smidt** [1]

## Abstract

$E(3)$-equivariant neural networks have proven to be effective in a wide range of 3D modeling tasks. A fundamental operation of such networks is the tensor product, which allows interaction between different feature types. Because this operation scales poorly, there has been considerable work towards accelerating this interaction. However, recently Xie et al. (2025) have pointed out that most speedups come from a reduction in expressivity rather than true algorithmic improvements on computing Clebsch-Gordan tensor products. A modification of Gaunt tensor product (Luo et al., 2024) can give a true asymptotic speedup but is incomplete and misses many interactions. In this work, we provide the first complete algorithm which truly provides asymptotic benefits Clebsch-Gordan tensor products. For full CGTP, our algorithm brings runtime complexity from the naive $O(L^6)$ to $O(L^4 \log^2 L)$, close to the lower bound of $O(L^4)$. We first show how generalizing fast Fourier based convolution naturally leads to the previously proposed Gaunt tensor product (Luo et al., 2024). To remedy antisymmetry issues, we generalize from scalar signals to irrep valued signals, giving us tensor spherical harmonics. We prove a generalized Gaunt formula for the tensor harmonics. Finally, we show that we only need up to vector valued signals to recover the missing interactions of Gaunt tensor product. An implementation of the tensor spherical harmonics based on `e3nn-jax` can be found here

## 1. Introduction

Symmetries are present in many complex physical systems, and incorporating them in neural networks can significantly improve both learning efficiency and robustness (Batzner et al., 2022; Rackers et al., 2023; Frey et al., 2023; Owen et al., 2024). As a result, considerable effort has been dedicated to develope $E(3)$-equivariant neural networks (E(3)NNs) (Thomas et al., 2018; Weiler et al., 2018; Kondor, 2018; Kondor et al., 2018). E(3)NNs have delivered strong performance across a wide range of scientific applications, including molecular force fields (Batzner et al., 2022; Musaelian et al., 2023; Batatia et al., 2022), catalyst discovery (Liao & Smidt, 2023), generative models (Hoogeboom et al., 2022), charge density prediction (Fu et al., 2024), and protein structure prediction (Jumper et al., 2021; Lee et al., 2022).

The group $E(3)$ consists of all rotations, translations and reflections in 3 dimensions; we say a model is $E(3)$-equivariant if it satisfies:

$$f(g \cdot x) = g \cdot f(x) \quad \forall g \in E(3), x \in X. \tag{1}$$

The way a feature transforms under a group is called a group representation. For most groups of interest, a representations can be broken into smaller pieces called irreducible representations (irreps). See Appendix B for a more in depth discussion. Hence, irreps are well studied objects in group theory and play a prominent role in many equivariant networks.

It is useful to choose a basis where any representation is explicitly a direct sum of irreps. This is done in popular $E(3)$ equivariant frameworks (Geiger & Smidt, 2022; Unke & Maennel, 2024; NVIDIA Corporation, 2025). Such a basis makes it easy to construct equivariant linear layers since Schur's Lemma tells us only irreps of the same type can mix. In order to interact irreps of different types, we can take a tensor product which is effectively an outer-product. However, the resulting representation is no longer explicitly a direct sum of irreps. The Clebsch-Gordan (CG) coefficients[1] (Varshalovich et al., 1988) provide the change of basis to convert the resulting tensor product representation

---

[1]Department of EECS, Massachusetts Institute of Technology, Cambridge, MA 02139, USA [2]Department of CSE, Massachusetts Institute of Technology, Cambridge, MA 02139, USA. Correspondence to: YuQing Xie <xyuqing@mit.edu>, Tess Smidt <tsmidt@mit.edu>.

*Proceedings of the 43rd International Conference on Machine Learning*, Seoul, South Korea. PMLR 306, 2026. Copyright 2026 by the author(s).

---

[1]The values of the Clebsch-Gordan coefficients are also basis dependent. In this paper we stick to quantum mechanical con-

back into a direct sum of irreps. The combination of the outer product along with the CG change of basis is referred to as the Clebsch-Gordan tensor product (CGTP).

However, CGTP is slow, limiting our ability to scale up these models for larger systems. In the typical setting, it has been reported to have a time complexity of $\mathcal{O}(L^6)$ (Passaro & Zitnick, 2023) though it can be reduced to $\mathcal{O}(L^5)$ by considering additional sparsity (Cobb et al., 2021). As a result, significant effort to speed up equivariant networks has focused on tensor products.

Some recent works have explored abandoning tensor products altogether. For instance, one line of works have suggested scalarizing high order features before interaction (Cen et al., 2024; Frank et al., 2024; Aykent & Xia, 2025). However, tensor products are proven to be universal (Dym & Maron, 2021) and seem necessary in certain tasks (Cen et al., 2025). Further, they provide a way to encode high symmetry information as higher order features (Cen et al., 2024; Lin et al., 2026). Hence, accelerating tensor products is an important and necessary part for scalable equivariance.

In molecular systems, one set of inputs for CGTP is often derived from spherical harmonics of relative difference vectors. In this special case, Passaro & Zitnick (2023) showed a suitable rotation creates significant sparsity allowing for faster operations. Further, Li et al. (2025) reduces number of this type of tensor product needed from number of edges $\mathcal{O}(|\mathcal{E}|)$ to number of vertices $\mathcal{O}(|\mathcal{V}|)$, allowing for better scaling on dense graphs.

On the engineering side, there have been efforts to build specialized kernels to optimize performance on current hardware. These include openEquivariance, FlashTP, cuEquivariance, and B3 (Bharadwaj et al., 2025; Lee et al., 2025; NVIDIA Corporation, 2025; Kotak, 2025) which work for general tensor products.

On the algorithmic side there have been a number of alternatives proposed. One line of work proposes working in a Cartesian basis instead of a strictly irrep (also called spherical) basis (Shao et al., 2024; Zaverkin et al.). In this framework taking tensor products is cheap, but decomposing back into irreps, especially to construct equivariant linear layers, now becomes expensive. However for CGTP, while these methods may be faster for small $L$, they have poor asymptotic scaling.

There have also been alternative tensor product operations (TPOs) (Luo et al., 2024; Unke & Maennel, 2024) with better asymptotic scaling proposed to replace CGTP. Recently, Xie et al. (2025) pointed out that these alternatives are not truly tensor products and their speedups directly

come from reduced expressivity. Using these to simulate full CGTP would lead to the same $\mathcal{O}(L^5)$ scaling. However, Xie et al. (2025) noticed that there are asymptotically fast SH transform algorithms (Healy et al., 2003) which allow the Gaunt tensor product (GTP) (Luo et al., 2024) to achieve additional asymptotic runtime gains. These additional gains are the only truly algorithmic ones, not a result of losing expressivity. Unfortunately, GTP is incomplete and misses interactions such as cross products and cannot fully simulate CGTP.

In this paper, we present the first TPO which truly provides asymptotic gains and is complete. It can be used to simulate full CGTP in $\mathcal{O}(L^4 \log^2 L)$ time, close to the theoretical lower bound of $\mathcal{O}(L^4)$. We present our results in a way which is generalizable to tensor products on other compact Lie groups. In addition, we provide a generalized version of the Gaunt formula for tensor spherical harmonics, which may be useful in other scientific fields. Our main contributions are the following

- First truly asymptotically faster and complete tensor product operation

- A generalized Gaunt formula for tensor spherical harmonics

- Explicit connection of tensor products to group Fourier transforms, allowing generalization to other groups

- Drop-in replacement for Gaunt tensor product that is complete

We organize this paper as follows. In Section 2, we briefly cover the framework for analyzing tensor product operations introduced in Xie et al. (2025). In Section 3, we draw an explicit connection to group Fourier transforms, showing that the previously proposed Gaunt tensor product (Luo et al., 2024) is a natural result of attempting to generalize the ideas of fast Fourier based convolutions. In Section 4, we generalize from the usual scalar spherical harmonics to tensor spherical harmonics. We show how to use tensor spherical harmonics to produce a TPO and derive generalized Gaunt coefficients to analyze this operation. In Section 5, we use this generalized formula to prove that we only need up to vector signals to perform any tensor product interaction, calling it vector signal tensor product (VSTP). Finally, in Section 6, we summarize the asymptotic runtimes of our algorithm compared to other TPOs.

## 2. Analyzing tensor products

A variety of tensor product operations (TPOs) have been proposed to accelerate the standard CGTP (Unke & Maennel, 2024; Luo et al., 2024). These can be defined as the following.

---

ventions. Most equivariant frameworks convert to a real basis, changing the standard Clebsch-Gordan coefficients.

**Definition 2.1** (Tensor product operations). Let $X, Y, Z$ be vector spaces equipped with actions of $G$. We refer to any fixed equivariant bilinear $T : X \times Y \to Z$ as a tensor product operation.

Xie et al. (2025) then defined an measure of expressivity based for the TPOs. In fact, they show that all of the asymptotic speed gains of previously proposed TPOs are directly explained by the loss in expressivity. Recovering the lost expressivity would require multiple copies of these TPOs in parallel, resulting in asymptotic runtime equivalent to standard CGTP with all sparsity constraints.

However, it may be possible that certain CGTP interactions cannot be simulated no matter how many times we call a given TPO. For instance, GTP can never simulate cross products. (Xie et al., 2025) proposed a measure of inter-actibility to describe this issue. Their definition is equivalent to the following.

**Definition 2.2** (Interactability). Let $T : X \times Y \to Z$ be a tensor product operation. Let $(\ell_1, \ell_2, \ell_3)$ be a triple of irrep types and $V_{\ell_1}, V_{\ell_2}, V_{\ell_3}$ be the corresponding spaces. Denote by $\mathrm{Hom}_G(A, B)$ to be the set of equivariant linear maps from $A \to B$. If there exists a nonzero bilinearity $B : V_{\ell_1} \times V_{\ell_2} \to V_{\ell_3}$ of the form

$$B(v_1, v_2) = L_Z(T(L_X(\mathbf{v_1}), L_Y(\mathbf{v_1})))$$

where $L_X \in \mathrm{Hom}_G(V_{\ell_1}, X)$, $L_Y \in \mathrm{Hom}_G(V_{\ell_2}, Y)$, $L_Z \in \mathrm{Hom}_G(Z, V_{\ell_3})$, then we say $(\ell_1, \ell_2, \ell_3)$ is interactable under $T$.

Intuitively, this just says $(\ell_1, \ell_2, \ell_3)$ is interactable if we can use $T$ to form a $\ell_3$ type irrep from a pair of $\ell_1, \ell_2$ inputs. GTP only allows even interactions where $\ell_1 + \ell_2 + \ell_3$ is even. For example, cross product corresponds to $(1, 1, 1)$ which has odd sum and is not computable.

We can directly analyze interactability by using selection rules. Let the irreps in $X, Y, Z$ be labeled by the tuples $(\ell_X, c_X), (\ell_Y, c_Y)$ and $(\ell_Z, c_Z)$, where $\ell$ indicates the irrep type and $c$ is an index over its multiplicity. A **selection rule** for $T$ is a condition on the labels $(\ell_X, c_X), (\ell_Y, c_Y), (\ell_Z, c_Z)$ which must be satisfied for

$$T|_{(\ell_X, c_X), (\ell_Y, c_Y), (\ell_Z, c_Z)} : X^{(\ell_X, c_X)} \times Y^{(\ell_Y, c_Y)} \to Z^{(\ell_Z, c_Z)}$$

to be nonzero. If there is no choice of $(c_X, c_Y, c_Z)$ such that $(\ell_X, c_X), (\ell_Y, c_Y)$ and $(\ell_Z, c_Z)$ satisfy the selection rules for a given $T$, then $(\ell_1, \ell_2, \ell_3)$ is not interactable under $T$. Hence, analyzing selection rules helps formally characterize which interactions are excluded by a specific TPO.

For $SO(3)$ (and $SU(2)$), the triangle condition often appears as a selection rule.

**Definition 2.3** (Triangle condition). Let $(\ell_1, \ell_2, \ell_3)$ be a triplet. They satisfy the triangle condition if

$$\ell_1 \leq \ell_2 + \ell_3 \qquad \ell_2 \leq \ell_1 + \ell_3 \qquad \ell_3 \leq \ell_1 + \ell_2.$$

It is convenient to define the triangular delta as $\{\ell_1, \ell_2, \ell_3\} = 1$ if the above is satisfied and 0 otherwise.

In particular, for single irrep CGTP the triangle condition is the only selection rule[2]. For GTP, we say there is an additional selection rule that $\ell_1 + \ell_2 + \ell_3$ must be even.

## 3. Generalizing FFT convolutions

In this section, we show how Gaunt tensor product naturally arises from attempting to generalize the ideas of FFT convolution for use in tensor products. We do so by first looking at the natural generalization of Fourier transforms to compact nonabelian groups. There are then two natural ways to generalize the convolution theorem, either we consider group convolutions or we consider pointwise multiplication of group signals. The former establishes the connection between group equivariant convolution networks and irrep based equivariant networks, while the latter is promising for computing tensor products. However, group Fourier transforms have an unwieldy irrep multiplicity. By quotienting the group with an appropriate subgroup, we remove redundant irreps. In the case of $SO(3)$, this quotient manifold is the sphere, giving us the spherical harmonics and the proposed Gaunt tensor product.

### 3.1. Group Fourier Transforms

In this context, a periodic function can be regarded as a signal on $S^1$. One can endow the circle with a group structure by addition of polar angle modulo $2\pi$ to create the circle group $\mathbb{T} \cong SO(2)$. The group action on the circle then induces an action on circle signals $S^1 \to \mathbb{C}$, namely a rotation by angle $\theta'$ rotates the signal $f(\theta) \to f'(\theta) = f(\theta - \theta')$. The set of circular signals forms a vector space and we can ask how it decomposes into irreps of $\mathbb{T}$. These irreps correspond exactly to signals of the form $e^{in\theta}$, the standard Fourier basis. The Fourier coefficients are precisely the coefficients obtained when we transform to this irrep basis.

This connection between signals on a group and their irreps is generalized by the Peter Weyl theorem.

**Theorem 3.1** (Peter-Weyl Theorem). *Let $G$ be a compact topological group. Let $\Sigma$ contain exactly one representative irrep from each isomorphism class of irreps. For each irrep $\pi \in \Sigma$, denote by $D^{(\pi)}$ the corresponding matrix for the irrep written in an orthonormal basis.*

*Then the space of square-integrable functions $L^2(G)$ has an orthonormal basis consisting of*

$$\{\sqrt{d_\pi} D_{ij}^{(\pi)} | \pi \in \Sigma, \ 1 \leq i, j \leq d_\pi\}$$

In the case of $SO(3)$, this means an orthonormal basis for

---

[2]For multiple irreps we need to track the multiplicities

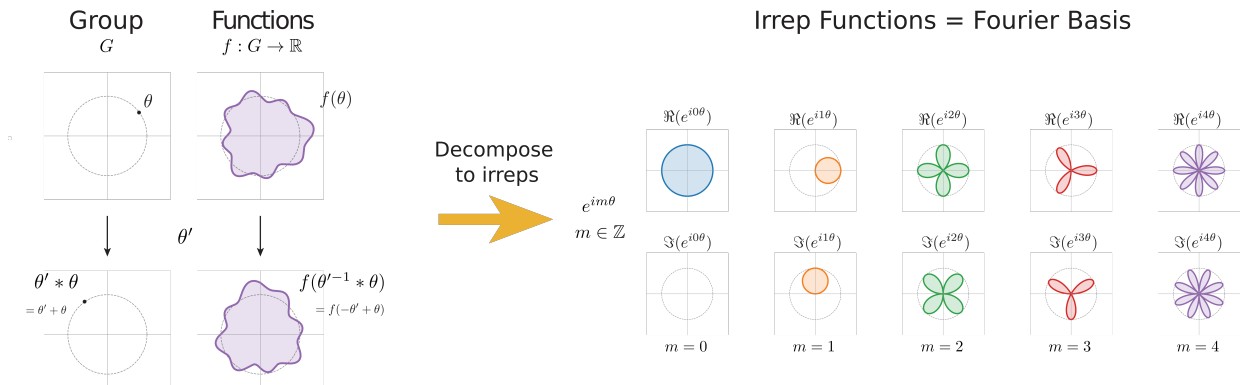

*Figure 1.* Fourier transforms from a group theoretic perspective. An action on a group induces an action on functions on the group. These functions form a vector space and hence this defines a group representation. We can decompose this representation into irreps, giving exactly the standard Fourier basis. The Peter-Weyl theorem generalizes this idea for compact Lie groups.

$SO(3)$ is given by the elements in the Wigner-D matrices scaled appropriately. We similarly then have generalized Fourier transform (GFT) coefficients obtained when transforming into this orthonormal basis. Due to orthonormality we can compute these as

$$M_{ij}^{(\pi)}(f) = \int_G \sqrt{d_\pi} f(g) D_{ij}^{(\pi)}(g) d\mu(g)$$

where $f : G \to \mathbb{C}$ is our signal and $\mu$ is the Haar measure.

### 3.2. Generalizing Convolution Theorems

For the usual Fourier transform of periodic functions, there are two convolution theorems. One states that convolution of two circular signals gives pointwise multiplication on the coefficients. The other states pointwise multiplication of two circular signals gives convolution on the coefficients. So there are two natural ways to generalize for compact groups. We can either perform convolution of two group signals or pointwise multiplication of two group signals.

Convolution of two group signals has been extensively explored in prior work and forms the basis of early equivariant models (Cohen & Welling, 2016; Cohen et al., 2018). The Fourier coefficients naturally organize into matrices, corresponding exactly into the matrices forming the orthonormal basis. The group convolution then becomes matrix multiplication of the coefficients (Kostelec & Rockmore, 2008; Cohen et al., 2018). In group convolution networks, we typically view one signal as our features and the other signal as a filter. In irrep focused frameworks such as `e3nn` (Geiger & Smidt, 2022), the Fourier coefficients of the features consist of $d_\ell$ copies of irreps of type $\ell$ and the coefficients of the filter correspond to invariant weights for an equivariant linear layer.

Less explored in equivariant literature is pointwise multiplication of two group signals. To understand the resulting action on the Fourier coefficients, one must understand how

the product of the basis matrices decompose. In the case of $SO(3)$, this is given by

$$
\begin{aligned}
D_{m_1 n_1}^{(\ell_1)}(g) & D_{m_2 n_2}^{(\ell_2)}(g) \\
&= \sum_{\ell_3} C_{\ell_1,m_1,\ell_2,m_2}^{\ell_3,m_1+m_2} C_{\ell_1,n_1,\ell_2,n_2}^{\ell_3,n_1+n_2} D_{m_1+m_2,n_1+n_2}^{(\ell_3)}(g)
\end{aligned}
$$

(2)

where here we use the typical convention that indices $m_i, n_i \in \{-\ell, -\ell+1, \ldots, \ell-1, \ell\}$. Note the appearance of the Clebsch-Gordan coefficients which is promising. This formula generalizes to other Lie groups where we replace the Wigner D with the corresponding irrep matrix and the Clebsch-Gordan coefficients with those for the other group.

Note however that for $SO(3)$ there are $d_\ell = 2\ell + 1$ copies of irreps of type $\ell$. This is rather unwieldy. Further, the corresponding GFT scales poorly. The corresponding Nyquist sampling theorem requires number of points scaling as $L^3$. Hence it infeasible to use this to speed up tensor products of a pair of single irreps and we must use multiple input irreps to possibly achieve better asymptotic runtime/expressivity ratios.

### 3.3. Reduction to Spherical Signals

Here, we describe how quotienting the $SO(3)$ manifold reduces the irrep multiplicity, giving us the sphere $S^2$. This lets us derive the Gaunt coefficients and shows how the Gaunt tensor product is a natural consequence of trying to generalize the key idea of FFT convolutions. This also lets us generalize the process for arbitrary Lie groups and gives the analog of the sphere.

Consider the $SO(2)$ subgroup generated by rotations about the $z$ axis. Then we can take the left cosets to create the quotient manifold $SO(3)/SO(2) \cong S^2$ where the equivalence to $S^2$ is a well known correspondence. To derive the spherical harmonics from a representation theory perspec-

tive, we realize that there are two natural group actions on $SO(3)$ itself. We can either perform group multiplication on the left or on the right. That is $g'$ acting on $g$ either gives $g \to g'g$ or $g \to gg'$. Depending on whether we choose left or right multiplication, the induced action on the $SO(3)$ signals changes and the $d_\ell$ copies of irrep $\ell$ are given either by the columns or rows respectively of the Wigner D matrices.

Next, we note that the indices $m, n$ describe exactly how much that component changes under $z$ rotations. That is if $g_z(\theta)$ is a rotation about $z$ by angle $\theta$, we have

$$D_{m,n}^{(\ell)}(gg_z(\theta)) = D_{m,n}^{(\ell)}(g)e^{in\theta}$$

$$D_{m,n}^{(\ell)}(g_z(\theta)g) = D_{m,n}^{(\ell)}(g)e^{im\theta}$$

If we quotient out by $z$ rotations acting on the right (giving left cosets), that means we need $n = 0$ to obtain functions which are invariant under $z$ rotations. This means $D_{m,0}^{(\ell)}(gg_z) = D_{m,0}^{(\ell)}(g)$ for all $g_z \in SO(2)$ and hence we expect $D_{m,0}^{(\ell)}$ to form a basis for $SO(3)/SO(2)$. It turns out that

$$D_{m,0}^{(\ell)}(g) = \sqrt{\frac{4\pi}{2\ell+1}} Y_\ell^{m,*}(g\hat{z})$$

so the Wigner D matrices indeed reduce to the spherical harmonics[3]. By substituting this relation into equation 2, we obtain

$$Y_{\ell_1}^{m_1,*}(g\hat{z})Y_{\ell_1}^{m_1,*}(g\hat{z})$$
$$= \sqrt{\frac{(2\ell_1+1)(2\ell_2+1)}{4\pi(2\ell_3+1)}} \times$$
$$\sum_{\ell_3} C_{\ell_1,m_1,\ell_2,m_2}^{\ell_3,m_1+m_2} C_{\ell_1,0,\ell_2,0}^{\ell_3,0} Y_{\ell_3}^{m_1+m_2,*}(g\hat{z}).$$

These are exactly the Gaunt coefficients (Gaunt, 1929).

Hence, we see that the Gaunt tensor product is what naturally arises when trying to exploit the pointwise multiplication analog of the convolution theorem for compact lie groups and then trying to form a quotient manifold to reduce the irrep multiplicity. It turns out $SO(2)$ is called a maximal torus and for general Lie groups, we can also quotient by a maximal torus to eliminate irrep multiplicity.

However, as also pointed by Xie et al. (2025) Gaunt tensor product suffers from an antisymmetry problem and many interactions such as cross products are not possible. In addition, the procedure of quotienting out the maximal torus does not guarantee every irrep is present in the corresponding harmonics. For example for $SU(2)$, the maximal torus is $U(1)$ and we still have $SU(2)/U(1) \cong S^2$. Hence we

---

[3]Throughout this paper, we use quantum mechanical conventions and our spherical harmonics are complex. This also facilitates analysis. The real spherical harmonics are related by a basis change and hence our main results still follow.

still only obtain the spherical harmonics and integer representations, missing all the half integer representations of $SU(2)$.

## 4. Generalizing Spherical Harmonics

To remedy the antisymmetry issue, we must generalize the spherical harmonics in a suitable way. In physics literature, there are two types of generalizations: spin weighted spherical harmonics (SWSH) and tensor spherical harmonics (TSH). The former were introduced by Newman & Penrose (1966) to study gravitational waves and also explored in the equivariant literature as a way to generalize spherical CNNs (Esteves et al., 2020). The latter is a decomposition of tensor valued signals on the sphere. The vector spherical harmonics in particular play an important role in many areas of physics (Barrera et al., 1985; Carrascal et al., 1991; Moses, 1974; Weinberg, 1994). The spin-weighted and tensor harmonics are intimately related and we believe both can be used to solve the antisymmetry issue (Thorne, 1980; Ledesma & Mewes, 2020). In this work, we will focus on tensor harmonics.

### 4.1. Tensor Spherical Harmonics

Suppose instead of a scalar signal, we instead have a tensor valued signal which transforms under some group representation. Since any representation can be decomposed as a sum of irreps, we can consider only irrep valued signals. Now when we act on such a signal, we must also transform the output. That is, any irrep $s$ signal $\mathbf{f}_s : S^2 \to \mathbb{R}^{2s+1}$ transforms as

$$\mathbf{f}^s(x) \to D^{(s)}(g)\mathbf{f}^s(g^{-1}x).$$

We can then similarly ask how to decompose the vector space of such signals into a direct sum of irreps and form an orthonormal basis.

We can interpret the transformation of irrep signals as a tensor product representation of a transform acting on the inputs, and a transform acting on the outputs. The spherical harmonics already tell us how to decompose the input transformation into irreps while the output representation is assumed to be an irrep. Therefore, to reduce the overall transformation into irreps, we can simply just use the Clebsch-Gordan decomposition. This leads to the following definition of tensor spherical harmonics.

**Definition 4.1** (Tensor spherical harmonics)**.** For integers $j, \ell, s, m$ where $|j - 1| \leq \ell \leq j + 1$ and $|m_j| \leq j$, we define the functions $\mathbf{Y}_{j,m_j}^{\ell,s} : S^2 \to \mathbb{R}^{2s+1}$ as

$$(\mathbf{Y}_{j,m_j}^{\ell,s}(\hat{\mathbf{r}}))_{m_s} = \sum_{m_\ell} C_{\ell,m_\ell,s,m_s}^{j,m_j} Y_\ell^{m_\ell}(\hat{\mathbf{r}}).$$

We refer to these functions as the tensor spherical harmonics (TSH).

In physics, $\ell$ is typically interpreted as orbital angular momentum, $s$ as spin, and $j$ as total angular momentum. Note that the TSH transforms as an irrep of type $j$.

*Remark* 4.2. We would like to point out that this choice for TSH is not unique. Ours is the most natural choice from a representation theory perspective, however there may be other choices that are natural for different contexts. For instance, $s = 1$ gives the vector spherical harmonics. However, the outputs of $\mathbf{Y}_{j,m}^{j-1,1}$ and $\mathbf{Y}_{j,m}^{j+1,1}$ are not purely perpendicular or tangent to the local tangent plane. Hence, one often sees an alternative basis consisting of purely radial, curl-free, and divergence free components.

### 4.2. Computing TSH transform

Because Definition 4.1 relates the TSH to regular scalar SHs, we can compute a forward TSH transform by performing multiple regular SH transforms and performing a Clebsch-Gordan decomposition on the resulting coefficients to obtain the TSH coefficients. For the reverse TSH transform, we can similarly use Clebsch-Gordan coefficients to transform into multiple regular SH coefficients and then just perform multiple reverse SH transforms.

Because we now compute multiple SH transforms and we compute a Clebsch-Gordan decomposition, the above procedure adds a complexity factor that depends on the chosen spin $s$. However as we shall see in Section 5, we only need up to $s = 1$ to be able to compute any tensor product.

### 4.3. Irrep signal tensor product

Given any two tensors, the most general bilinear interaction is a tensor product. Therefore, given any two irrep signals, the most general interaction is to perform a pointwise tensor product. The resulting signal output transforms as a tensor product representation. We can then use the Clebsch-Gordan decomposition to change to an irrep basis. In particular, if we have irrep signals $\mathbf{f}_{s_1}$ and $\mathbf{f}_{s_2}$ and desire a resulting irrep signal of type $s_3$, we can get

$$(\mathbf{f}^{s_1} \otimes \mathbf{f}^{s_2})_{m_3}^{s_3}(\hat{\mathbf{r}}) = \sum_{m_1,m_2} C_{s_1,m_1,s_2,m_2}^{s_3,m_3} \mathbf{f}_{m_1}^{s_1}(\hat{\mathbf{r}}) \mathbf{f}_{m_2}^{s_2}(\hat{\mathbf{r}}).$$

If $s_1 = s_2 = s_3 = 0$, the above operation is the usual pointwise multiplication of scalar signals.

Using this interaction, we can now define a general class of TPOs we call irrep signal tensor products (ISTPs). For any triplet $(s_1, s_2, s_3)$, we can perform the following

1. Interpret input irreps labeled by $(j_i, \ell_i)$ as coefficients for a set of TSHs $\mathbf{Y}_{j_i}^{\ell_i, s_i}$.

2. Do a reverse TSH transform to form a tensor signal

$$\mathbf{f}^{s_i} = \sum_{j_i, \ell_i, m_i} (\mathbf{x}_i)_{m_i}^{(j_i, \ell_i)} \mathbf{Y}_{j_i, m_i}^{\ell_i, s_i}.$$

3. Compute pointwise tensor products to obtain $\mathbf{f}^{s_3} = (\mathbf{f}^{s_1} \otimes \mathbf{f}^{s_2})^{s_3}$.

4. Finally, we perform a TSH transform on $\mathbf{f}^{s_3}$ to get corresponding TSH coefficients labeled by $(j_3, \ell_3)$.

Note that the $(0, 0, 0)$ version corresponds to Gaunt tensor product.

### 4.4. Generalized Gaunt formula

To analyze the ISTP procedure, we need a formula to decompose 2 TSHs.

**Theorem 4.3.** *We have the following decomposition of tensor spherical harmonics*

$$(\mathbf{Y}_{j_1,m_1}^{\ell_1,s_1} \otimes \mathbf{Y}_{j_2,m_2}^{\ell_2,s_2})_{s_3}$$
$$= \sqrt{\frac{(2j_1+1)(2j_2+1)(2\ell_1+1)(2\ell_2+1)(2s_3+1)}{4\pi}}$$
$$\sum_{j_3,\ell_3,m_3} \begin{Bmatrix} j_1 & \ell_1 & s_1 \\ j_2 & \ell_2 & s_2 \\ j_3 & \ell_3 & s_3 \end{Bmatrix} C_{\ell_1,0,\ell_2,0}^{\ell,0}$$
$$C_{j_1,m_1,j_2,m_2}^{j_3,m_3} \mathbf{Y}_{j_3,m_3}^{\ell_3,s_3}.$$

Here, the $3 \times 3$ curly brace denotes the Wigner 9j symbol defined in Definition F.3.

*Proof sketch.* To compute the signal at each point, we first couple our TSH coefficient of type $j_i$ with a spherical harmonic of type $\ell_i$ to form a resulting $s_i$. We then couple $s_1, s_2$ to form the final $s_3$. This is a specific choice to couple 4 irreps of type $\ell_1, j_1, \ell_2, j_2$. However, we can alternatively couple $\ell_1, \ell_2$ to form $\ell_3$ and $j_1, j_2$ to form $j_3$ before coupling $\ell_3, j_3$ to form $s_3$. The Wigner 9j gives the change of basis between these coupling orders. In the latter, the $\ell_1, \ell_2$ coupling gives a $\ell_3$ spherical harmonic and the $j_1, j_2$ coupling gives the desired CGTP coefficients. The last $\ell_3, j_3$ coupling is exactly a TSH coupling giving the final $\mathbf{Y}_{\ell_3,s_3}^{j_3,m_3}$.

Full proof is in Section F.1. $\qquad\square$

*Remark* 4.4. To the best of our knowledge, this general formula and our derivation method is novel. Versions for $s \leq 1$ can be found in Varshalovich et al. (1988).

## 5. Vector Signals is All You Need

In this section, we prove only up to $s = 1$ is needed to perform all possible tensor product interactions. In particular, we focus on the $(1, 1, 1)$ ISTP which is equivalent to performing cross products of vector spherical signals (up to a factor of $\sqrt{2}$). We call this vector signal tensor product (VSTP). We then derive the corresponding selection rules

and then prove one always has an interaction. Finally, we show a constant number of VSTPs can be used to simulate computing full CGTP of a pair of irreps.

## 5.1. Vector signal tensor product

Here, we consider $s = 1$ signals, which corresponds to vector signals on the sphere. Inspecting the Clebsch-Gordan coefficients, one can show that

$$(\mathbf{f}^{(1)} \otimes \mathbf{g}^{(1)})_{m_3}^{(1)} = -\frac{1}{\sqrt{2}}(\mathbf{f}^{(1)} \times \mathbf{g}^{(1)})_{m_3}$$

so the interaction is proportional to taking a cross product.

## 5.2. Selection rules and completeness

Using Theorem 4.3, we can derive selection rules for VSTP.

**Theorem 5.1** (Selection rules for VSTP). *Consider irrep labels $j_1, \ell_1, j_2, \ell_2, j_3, \ell_3$. The corresponding interaction is nonzero if and only if the following are satisfied:*

1. $\{j_1, \ell_1, 1\} = \{j_2, \ell_2, 1\} = \{j_3, \ell_3, 1\} = 1$

2. $\{j_1, j_2, j_3\} = 1$

3. $\{\ell_1, \ell_2, \ell_3\} = 1$

4. $\ell_1 + \ell_2 + \ell_3$ *is even*

5. *There is no choice of distinct $a, b, c$ such that $j_a = \ell_a$ and $(j_b, \ell_b) = (j_c, \ell_c)$*

*Proof sketch.* Rules 1-3 follow from the fact that the triangle condition must be satisfied in the rows and columns of the Wigner 9j symbol. Rule 5 follows from the (anti)symmetry relation of the Wigner 9j under odd row or column permutations. Rule 4 follows from the even selection rule of the $C_{\ell_1,0,\ell_2,0}^{\ell_3,0}$ coefficient.

To show these are always nonzero, one looks at the explicit expressions of the Wigner 9j symbols documented in Varshalovich et al. (1988). Tedious inspection of these coefficients then show that they only become zero when it fails the selection rules 1-3 and 5. Finally, it is known that $C_{\ell_1,0,\ell_2,0}^{\ell,0}$ is nonzero if and only if it satisfies rule 4 (Raynal, 1978).

Full proof is in Section F.2. □

In contrast to GTP, VSTP selection rules allow all the possible paths except for multiplication of scalars.

**Theorem 5.2.** *Suppose $\{j_1, j_2, j_3\} = 1$ and the $j's$ are not all 0. Then $(j_1, j_2, j_3)$ is interactable under a VSTP of sufficiently high degree.*

*Proof sketch.* We perform tedious casework on the possible $\ell_i$ for each $j_i$ and show we can always find some $\ell_1, \ell_2, \ell_3$ such that $j_1, \ell_1, j_2, \ell_2, j_3, \ell_3$ satisfies the selection rules of Theorem 5.1.

Full proof is in Section F.3. □

## 5.3. Simulating CGTP with VSTP

**Corollary 5.3.** *Suppose we have a pair of inputs of irrep types $j_1, j_2$. We can compute $(\mathbf{x}^{(j_1)} \otimes \mathbf{x}^{(j_2)})^{(j_3)}$ with a constant number of VSTPs.*

*Proof.* If $j_1 = j_2 = j_3 = 0$, then they are all scalars and we multiply using no VSTPs.

Each input $j_i$ has up to 3 possible $\ell_i$ it can be paired with. We simply compute a VSTP for each possible combination $j_1, \ell_1$ and $j_2, \ell_2$. There are at most $3 \cdot 3 = 9$ such combinations. By Theorem 5.2, there is at least one $j_3, \ell_3$ that works with one of the the computed VSTPs. □

# 6. Asymptotic runtimes and expressivity

## 6.1. Expressivity runtime tradeoffs

We analyzed the asymptotic runtimes and expressivity respectively in Appendix D and Appendix E of ISTP and VSTP. The results are summarized in Table 1. For expressivity, we assume we are using the various tensor products to construct bilinearities from space $X = Y$ transforming as $(0 \oplus \ldots \oplus L)$ to space $Z$ transforming as $(0 \oplus \ldots \oplus 2L)$.

*Table 1.* Asymptotic runtimes and expressivity of various tensor product implementations. Note that the fast spherical transform algorithm by Healy et al. (2003) gives the best runtime/expressivity tradeoff ratios.

| Tensor Product | Expressivity | Runtime | Runtime / Expressivity |
|---|---|---|---|
| CGTP (Naive) | $\mathcal{O}(L^3)$ | $\mathcal{O}(L^6)$ | $\mathcal{O}(L^3)$ |
| CGTP (Sparse) | $\mathcal{O}(L^3)$ | $\mathcal{O}(L^5)$ | $\mathcal{O}(L^2)$ |
| GTP (Fourier) | $\mathcal{O}(L)$ | $\mathcal{O}(L^3)$ | $\mathcal{O}(L^2)$ |
| GTP (Grid) | $\mathcal{O}(L)$ | $\mathcal{O}(L^3)$ | $\mathcal{O}(L^2)$ |
| GTP (Healy) | $\mathcal{O}(L)$ | $\mathcal{O}(L^2 \log^2 L)$ | $\mathcal{O}(L \log^2 L)$ |
| VSTP (Grid) | $\mathcal{O}(L)$ | $\mathcal{O}(L^3)$ | $\mathcal{O}(L^2)$ |
| VSTP (Healy) | $\mathcal{O}(L)$ | $\mathcal{O}(L^2 \log^2 L)$ | $\mathcal{O}(L \log^2 L)$ |
| ISTP (Grid) | $\mathcal{O}(\tilde{s}L)$ | $\mathcal{O}(\tilde{s}^2 L^2 + \tilde{s}L^3)$ | $\mathcal{O}(\tilde{s}L + L^2)$ |
| ISTP (S2FFT) | $\mathcal{O}(\tilde{s}L)$ | $\mathcal{O}(\tilde{s}^2 L^2 + \tilde{s}L^2 \log^2 L)$ | $\mathcal{O}(\tilde{s}L + L \log^2 L)$ |

Note that because $s = 1$ is a finite cutoff, VSTP has the same asymptotics as GTP.

## 6.2. CGTP simulation

By Corollary 5.3, we can fully simulate CGTP for a pair of irreps with a constant number of VSTPs. In the setting where both inputs are of type $(0 \oplus \ldots \oplus L)$, there are $(L + 1)^2 = \mathcal{O}(L^2)$ ways to pair input irreps. Hence, we need $\mathcal{O}(L^2)$ VSTP calls to simulate the corresponding CGTP.

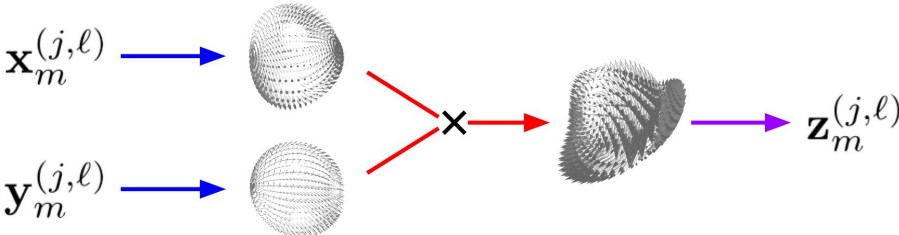

*Figure 2.* Schematic of the process in taking a vector signal tensor product. We interpret input irreps as vector SH coefficients to create vector spherical signals. We then take pointwise cross products of the two signals to create a new signal which we decompose back into vector SH coefficients.

Since each VSTP call runs in $\mathcal{O}(L^2 \log^2 L)$ time, this gives $\mathcal{O}(L^4 \log^2 L)$ runtime to fully simulate CGTP.

## 7. Experiments

We implemented the TSH and VSTP on top of the `e3nn-jax` library. The code can be found at https://github.com/atomicarchitects/tensor-spherical-harmonics. Note that `e3nn-jax` does not have the fast ($\mathcal{O}(L^2 \log^2 L)$) SH transform so our VSTP implementation does not achieve the theoretical $\mathcal{O}(L^2 \log^2)$ runtime. In fact, we are not aware of any python implementations of asymptotically fast SH transforms. Nonetheless, we can perform some simple sanity checks.

### 7.1. Classifying 3D Tetris Pieces

Consider the simple task of classifying 8 different 3D Tetris-like pieces, shown in 3a. Note that the first two pieces are **chiral**, that is they are non-superimposable mirror reflections of each other. Given a randomly oriented 3D structure, our task is to predict which of the 8 tetris pieces corresponds to. It was previously shown that Gaunt tensor product is unable to distinguish the chiral pieces and can only achieve 80% accuracy (Xie et al., 2025).

We follow the same setup as (Xie et al., 2025) described in more detail in Appendix G. We simply replace GTP with VSTP. The network architecture is almost identical to that of NequIP (Batzner et al., 2022).

As shown in 3b, the network with VSTP is able to perfectly solve the task as expected from the completeness shown by our interactability theorem.

### 7.2. Microbenchmarks

Following the setup in (Xie et al., 2025), we also microbenchmark our VSTP implementation against the GTP and CGTP implementation. As expected from theory, VSTP is essentially a constant factor slower than GTP. It is faster than CGTP for $\ell \geq 3$ though less expressive. Empirical

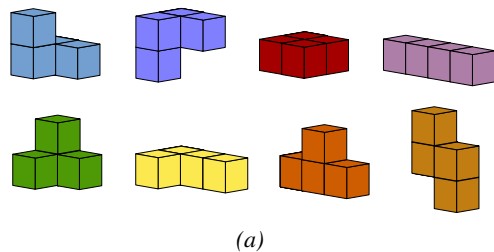

*(a)*

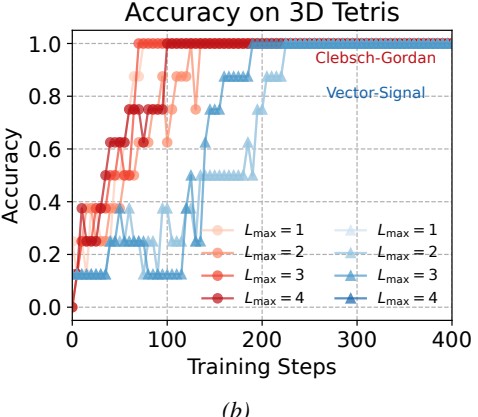

*(b)*

*Figure 3.* (a) The 8 different 3D Tetris pieces, with the first two pieces being mirror images of each other. (b) Training curves of networks trained with different tensor products on the 3D Tetris task. The maximum $L$ is varied from 1 to 4. Both the Clebsch-Gordan and VSTP networks attain 100% accuracy.

tests in larger models are necessary to determine whether this lost expressivity really impacts performance.

### 7.3. Symbolics checks

Finally, we also implemented a symbolic check of Theorem 4.3. This was done using `sympy`. We symbolically expanded the left hand side of the relation and the right hand side purely in terms of regular spherical harmonics. We then take the difference and also expand the spherical harmonics into its components. We simplify and check that the final expression is indeed 0.

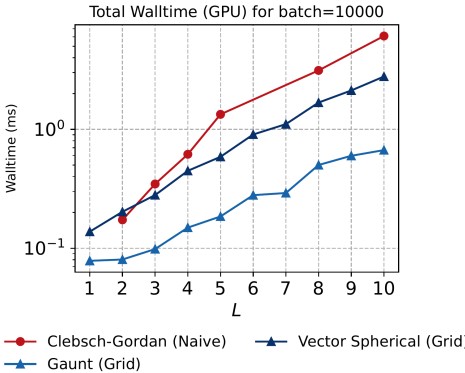

*Figure 4.* Microbenchmarks of VSTP against standard CGTP and GTP. We see VSTP has similar scaling as GTP differing by a constant factor. CGTP walltime increases faster than both as expected.

## 8. Limitations

While our work presents the first asymptotic improvements for computing full CGTP, it is not practical for the $L$'s currently used in equivariant networks. In the spherical harmonic transform, asymptotically fast $\mathcal{O}(L^2 \log^2 L)$ versions often suffer from numerical stability compared to $\mathcal{O}(L^3)$ versions. In addition, runtime benefits for fast SH transforms are often only achieved for $L \sim 1000$, far higher than any $L$ currently used in E(3)NNs.

However, there are many domains which use extremely high $L$. For instance, the Earth Gravitation Model uses $L \sim 2000$ (Pavlis et al., 2008) and planetary topography models go up to $L \sim 40,000$ (Rexer & Hirt, 2015). It may be possible that the ideas in this paper will be useful in the far future for such domains.

Lastly, it is possible VSTP with $\mathcal{O}(L^3)$ SH transforms is viable on current equivariant networks. It would be quite similar to the already viable GTP with a constant factor increase in expressivity, no interactibility problems, but also constant factor speed decrease. However, we have not robustly tested our preliminary implementation. In addition, many common benchmarks compute quantities such as forces or energies which can already be formed from the interactions of GTP. Hence, the lost interactibility of GTP may not be an issue in such tasks.

## 9. Conclusion

In this work, we investigated the use of spherical signals to perform Clebsch-Gordan tensor products. We first explicitly drew connections to group Fourier transforms, showing how a natural extension of FFT ideas to compute convolutions leads to GTP. We then showed that by generalizing to tensor spherical signals, we can obtain new types of interactions

and circumvent antisymmetry issues. We derived a generalized Gaunt formula for tensor harmonics under these interactions. Next, we showed only up to vector spherical harmonics is necessary to fully simulate CGTP. Finally, we analyzed the asymptotics of our proposed VSTP, sharing the same benefits as GTP from fast SH transforms but not suffering from interactibility issues.

For future work, we would like to robustly test the viability of VSTP in actual E(3)NNs. This may require careful initialization and normalization which our generalized Gaunt formula can help analyze. Further, we believe it will be fruitful to explore spin-weighted spherical harmonics, the other possible generalization of scalar SH.

## Acknowledgments

YuQing Xie, Ameya Daigavane, and Mit Kotak were supported by the NSF Graduate Research Fellowship program under Grant No. DGE-1745302. We were also supported by the U.S. Department of Energy, Office of Science, under Award No. DE-SC0026242, Hierarchical Representations of Complex Physical Systems with Euclidean Neural Networks and the National Science Foundation under Cooperative Agreement PHY-2019786 (The NSF AI Institute for Artificial Intelligence and Fundamental Interactions).

## Impact Statement

This paper presents work whose goal is to advance the field of Machine Learning. There are many potential societal consequences of our work, none which we feel must be specifically highlighted here.

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

## Table of Contents

## A. Notation

Here, we present the notation we use throughout this paper and the typical variable names.

*Table 2.* Notation used throughout this paper

| | |
|---|---|
| $SO(n)$ | Group of rotations in $n$-dimensional space |
| $O(n)$ | Group of rotations and inversion in $n$-dimensional space |
| $S^2$ | The 2-sphere, surface defined by $x^2 + y^2 + z^2 = 1$ |
| $Y_\ell^m$ | Spherical harmonic function of degree $\ell$ and order $m$ |
| $\mathbf{Y}_\ell$ | The collection of spherical harmonic functions of degree $\ell$ for all orders $m$ |
| $\mathbf{Y}_{j,m}^{\ell,s}$ | Tensor spherical harmonic function |
| $\mathbf{Y}_j^{\ell,s}$ | Collection of tensor spherical harmonic functions for all $m$ |
| $\oplus$ | Denotes a direct sum |
| $\otimes$ | Denotes a tensor product |
| $\times$ | Denotes a Cartesian product of spaces and also denotes cross products |
| $\mathcal{O}$ | Big O notation |
| ToSphere | Function which takes in spherical harmonic coefficients consisting of single copies of irrep up to some cutoff $L$ and converts it into a spherical signal $f : S^2 \to \mathbb{R}$ |
| FromSphere | Function takes in a spherical signal $f : S^2 \to \mathbb{R}$ and converts it to spherical harmonic coefficients consisting of single copies of irrep up to some cutoff $L$ |

*Table 3.* Commonly used meanings of symbols

| | |
|---|---|
| $G$ | Denotes a group |
| $\rho(g)$ | Representation of a group |
| $A$ | First input space of a constructed bilinearity |
| $B$ | Second input space of a constructed bilinearity |
| $C$ | Output space of a constructed bilinearity |
| $X$ | First input space of a tensor product operation (fixed equivariant bilinearity) |
| $Y$ | Second input space of a tensor product operation (fixed equivariant bilinearity) |
| $Z$ | Output space of a tensor product operation (fixed equivariant bilinearity) |
| $T$ | Tensor product operation (fixed equivariant bilinearity $X \times Y \to Z$) |
| $\ell$ | Typically used to denote irrep type for $SO(3)$. For spherical signals, used instead to denote spherical harmonic degree which naturally indexes multiplicities of irrep types for VSTP/ISTPs |
| $c$ | Indexes multiplicities of an irrep type in given space (channel) |
| $j$ | Used instead of $\ell$ to denote irrep type for VSTP and general ISTPs |
| $s$ | Denotes irrep type of spherical signal (ie. our signal is a map $f : S^2 \to \mathbb{R}^{2s+1}$) |

## B. Irreducible Representations of $E(3)$

A representation $\rho$ of a group $G$ maps each group element $g$ to a bijective linear transformation $\rho(g) \in \mathrm{GL}(V)$, where $V$ is some vector space. Representations must preserve the group multiplication property:

$$\rho(g \cdot h) = \rho(g) \circ \rho(h) \quad \forall g, h \in G \tag{3}$$

Thus, the representation $\rho$ defines a group action on a vector space $V$. The dimension of the representation $\rho$ is simply defined as the dimension of the vector space $V$.

There may be subspaces $W \subset V$ which are left invariant under actions of $\rho(g)$ for all $g \in G$. If this is the case, then restricting to $W$ also gives a representation $\rho|_W(g) \in \mathrm{GL}(W)$. If there is no nontrivial $W$, then we say the representation $\rho$ is an irreducible representation (irrep).

To build $E(3)$-equivariant neural networks, the irreducible representations of $E(3)$ play a key role. Because $E(3)$ is not a compact group, the usual approach has been to consider irreducible representations of the group $O(3)$ of 3D rotations + inversion, and compose them with the representation in which translations act as the identity:

$$\rho(R, T) = \rho'(R) \tag{4}$$

This is why translations are often handled in $E(3)$-equivariant neural networks by centering the system or only using relative vectors. Further, $O(3) \cong SO(3) \times \mathbb{Z}_2$. Hence, we label $O(3)$ irreps by $SO(3)$ irrep type and $\mathbb{Z}_2$ irrep type. The latter can be thought of as parity, whether we flip sign under inversion. Therefore $O(3)$ irreps are written as $\ell e$ or $\ell o$ where $\ell$ is a nonnegative integer and $e, o$ is even or odd parity corresponding to invariance under inversion or sign flip under inversion.

The 'scalar' representation $\rho_{\text{scalar}}$ representation of $SO(3)$ is defined as:

$$\rho_{\text{scalar}}(R) = \text{id} \quad \forall R \in SO(3) \tag{5}$$

and is of dimension $1$ over $V = \mathbb{R}$. Elements of $\mathbb{R}$ are unchanged by any rotation $R$. We call such elements 'scalars' to indicate that they transform under the 'scalar' representation of $SO(3)$. An example of a 'scalar' element could be mass of an object, which does not change under rotation of coordinate frames.

Let $T(R) \in \mathbb{R}^{3 \times 3}$ be the rotation matrix corresponding to a rotation $R \in SO(3)$. Then, the 'vector' representation of $SO(3)$ is defined as:

$$\rho_{\text{vector}}(R) = T(R) \quad \forall R \in SO(3) \tag{6}$$

and is of dimension $3$ over $V = \mathbb{R}^3$. The name arises from the way vectors in $\mathbb{R}^3$ transform under a rotation of the coordinate frame. We call such elements 'vectors' to indicate that they transform under the 'vector' representation of $SO(3)$. For example, the velocity and position of an object in a certain coordinate frame are 'vectors'.

Weyl's theorem for the Lie group $SO(3)$ states that all finite-dimensional representations of $SO(3)$ are equivalent to direct sums of irreducible representations. The irreducible representations of $SO(3)$ are indexed by an integer $\ell \geq 0$, with dimension $2\ell + 1$. $\ell = 0$ corresponds to the 'scalar' representation, while $\ell = 1$ corresponds to the 'vector' representation above. We will often use $m$, where $-\ell \leq m \leq \ell$, to index of each of the $2\ell + 1$ components.

We say that a quantity $\mathbf{x} \in \mathbb{R}^{2\ell+1}$ is a $\ell$ irrep, if it transforms as the irreducible representation ('irrep') of $SO(3)$ indexed by $\ell$. If $\mathbf{x}_1$ is a $\ell_1$ irrep and $\mathbf{x}_2$ is an $\ell_2$ irrep, we say that $(\mathbf{x}_1, \mathbf{x}_2)$ is a direct sum of $\ell_1$ and $\ell_2$ irreps, which we call a $(\ell_1, \ell_2)$ 'rep'. Weyl's theorem states that all reps are a direct sum of $\ell_i$ irreps, possibly with repeats over $\ell_i$: $\mathbf{x} = \oplus_{\ell_i} \mathbf{x}^{(\ell_i)}$. The multiplicity of an irrep in a rep is exactly the number of repeats.

An important lemma for constructing equivariant linear layer is Schur's lemma (Dresselhaus et al., 2007).

**Lemma B.1** (Schur's Lemma). *Suppose $V_1, V_2$ are irreps of a Lie group over any algebraically closed field (such as $SO(3)$). Let $\phi : V_1 \to V_2$ be an equivariant linear map.*

*Then $\phi$ is either $0$ or an isomorphism.*

*Further, if $V_1 = V_2$ then $\phi$ is a multiple of identity.*

*Finally for any two $\phi_1, \phi_2 : V_1 \to V_2$ we must have $\phi_1 = \lambda \phi_2$.*

This tells us that to construct equivariant linear layers between reps written as a direct sum of irreps, we can only have weights between input and output irreps of the same type and that those weights must be tied together so they give multiples of the identity transformation.

## C. Spherical Harmonics

The spherical harmonics are intimately connected to the representations of $SO(3)$ and play a key role in the Gaunt tensor product.

We define the spherical coordinates $(r, \theta, \varphi)$ as:

$$\begin{bmatrix} x \\ y \\ z \end{bmatrix} = \begin{bmatrix} r \sin\theta \cos\varphi \\ r \sin\theta \sin\varphi \\ r \cos\theta \end{bmatrix} \tag{7}$$

for $\theta \in [0, \pi), \varphi \in [0, 2\pi)$.

The spherical harmonics $Y_{\ell,}^m$ are a set of functions $S^2 \to \mathbb{R}$ indexed by $(\ell, m)$, where again $\ell \geq 0, -\ell \leq m \leq \ell$. Here, $S^2 = \{(r, \theta, \phi) \mid r = 1\}$ denotes the unit sphere.

Indeed, as suggested by the notation, the spherical harmonics are closely related to the irreducible representations of $SO(3)$. Let $\mathbf{Y}_\ell$ be the concatenation of all $Y_\ell^m$ over all $m$ for a given $\ell$:

$$\mathbf{Y}_\ell(\theta, \phi) = \begin{bmatrix} Y_{\ell,-\ell}(\theta, \phi) \\ Y_{\ell,-\ell+1}(\theta, \phi) \\ \dots \\ Y_{\ell,\ell}(\theta, \phi) \end{bmatrix} \tag{8}$$

When we transform the inputs to $Y_\ell(\theta, \phi)$, the output transforms as a $\ell$ irrep.

The spherical harmonics satisfy orthogonality conditions:

$$\int_{S^2} Y_{\ell_1}^{m_1,*} \cdot Y_{\ell_2}^{m_2} \, dS^2 = \delta_{\ell_1 \ell_2} \delta_{m_1 m_2} \tag{9}$$

where:

$$\int_{S^2} f \cdot g \, dS^2 = \int_{\theta=0}^{\pi} \int_{\varphi=0}^{2\pi} f(\theta, \varphi) g(\theta, \varphi) \sin \theta d\theta d\varphi \tag{10}$$

The orthogonality property allows us to treat the spherical harmonics as a basis for functions on $S^2$. We can linearly combine the spherical harmonics using irreps to approximate arbitrary functions on the sphere. Given a $(0, 1, \dots, L)$ rep $\mathbf{x} = (\mathbf{x}^{(0)}, \mathbf{x}^{(1)}, \dots, \mathbf{x}^{(L)})$, we can associate the function $f_{\mathbf{x}} : S^2 \to \mathbb{R}$ as:

$$f_{\mathbf{x}}(\theta, \varphi) = \sum_{\ell=0}^{L} \sum_{m=-\ell}^{\ell} \mathbf{x}_m^{(\ell)} Y_{\ell,m}(\theta, \varphi) \tag{11}$$

The function $f_{\mathbf{x}}$ is uniquely determined by $\mathbf{x}$. In particular, by the orthogonality of the spherical harmonics (Equation 9), we can recover the $\mathbf{x}_m^{(\ell)}$ component:

$$\mathbf{x}_m^{(\ell)} = \int_{S^2} f_{\mathbf{x}} \cdot Y_\ell^m \, dS^2 \tag{12}$$

## D. Runtime Analysis

Here, we provide a detailed asymptotic analysis of runtimes for different tensor products. We consider 3 different settings.

- **Single Input, Single Output (SISO)**:
  Here we are computing only one path $[\ell_1, \ell_2, \ell_3]$ where $\ell_i \in \mathcal{O}(L)$.
  $$\ell_1 \times \ell_2 \to \ell_3$$

- **Single Input, Multiple Output (SIMO)**:
  Here we fix $\ell_1, \ell_3$ but allow all possible irreps generated by the respective tensor products.
  $$\ell_1 \times \ell_2 \to Z$$

- **Multiple Input, Multiple Output (MIMO)**:
  Here we only bound the $L$ that the tensor products use but allow full capacity for the input and output irreps. In the case of CGTP, we can have an arbitrary number of copies of each irrep but we assume we only use single copies of each irrep in the input.
  $$X \times Y \to Z$$

In the SISO and SIMO settings, the asymptotic runtimes of different tensor products are directly comparable. However, in the MIMO setting, we lose expressivity in some tensor products. This is discussed more in Appendix E. Note the MIMO setting is what one would typically want to use in practice.

### D.1. Irrep Signal Tensor Products

Suppose we want to interpret our input irreps as coefficients for irrep signals of type $s$. Then for any given irrep of type $j$, it can be coefficients of any $\mathbf{Y}_{j,\ell,s}^{m_j}$ where $|j - s| \leq \ell \leq j + s$. At most there can be up to $2s + 1$ choices of $\ell$. We can flip this condition and see that we also have $|\ell - s| \leq j \leq \ell + s$ so for given $\ell$ there are only up to $2s + 1$ choices of input irrep $j$ which work. Hence, if we use scalar SH up to degree $L$, we can input $\mathcal{O}(sL)$ irreps into our signal.

Next, for encoding we can first convert the input irreps into coefficients of scalar spherical harmonics. Using the definition of our tensor harmonics, we have

$$\sum_{m_j} \mathbf{x}_{m_j}^{(j,\ell)} (\mathbf{Y}_{j,m_j}^{\ell,s})_{m_s}(\hat{\mathbf{r}}) = \sum_{m_j} \sum_{m_\ell} C_{\ell,m_\ell,s,m_s}^{j,m_j} \mathbf{x}_{m_j}^{(j,\ell)} Y_\ell^{m_\ell}(\hat{\mathbf{r}})$$

$$= \sum_{m_\ell} \left( \sum_{m_j} C_{\ell,m_\ell,s,m_s}^{j,m_j} \mathbf{x}_{m_j}^{(j,\ell)} \right) Y_\ell^{m_\ell}(\hat{\mathbf{r}})$$

$$= \sum_{m_\ell} A_{m_s,m_\ell}^{(j,\ell)} Y_\ell^{m_\ell}(\hat{\mathbf{r}})$$

where $A_{m_s,m_\ell}^{(j,\ell)}$ are coefficients of the scalar SH. To compute these coefficients, naively we have to perform a summation over $m_j$ taking $\mathcal{O}(j)$ for each pair of $m_\ell, m_s$ giving runtime of $\mathcal{O}(js\ell)$. However, leveraging sparsity reduces this to $\mathcal{O}(s\ell)$. Finally, we can use the same fast SH transforms (Healy et al., 2003) to convert into spherical signals for each component which takes $\mathcal{O}(\ell^2 \log \ell)$ for single components giving $\mathcal{O}(s\ell^2 \log \ell) = \mathcal{O}(sL^2 \log L)$ for all components.

If we now allow all irreps, we need to compute $\mathcal{O}(sL)$ coefficients for $\mathcal{O}(s^2L^2)$ time. Next, we can compute

$$B_{m_s,m_\ell}^{\ell} = \sum_j A_{m_s,m_\ell}^{(j,\ell)}.$$

There are $\mathcal{O}(s)$ values of valid $j$ for given $\ell$ and each $A$ has $\mathcal{O}(sL)$ components for $\mathcal{O}(s^2L^2)$ time to compute the $B$'s. Finally, for each $m_s$ we can use the $B$'s to compute a fast reverse SH transform for the signal values for a runtime of $\mathcal{O}(L^2 \log^2 L)$ for each component. Hence we have $\mathcal{O}(sL^2 \log^2 L)$ to compute all components. This gives $\mathcal{O}(s^2L^2 + sL^2 \log^2 L)$ runtime for converting all input irreps into an irrep signal on a grid.

Next, recall our grid has $\mathcal{O}(L^2)$ points. At each point, we perform a CGTP operation and extract an irrep $s_3$ from $s_1 \otimes s_2$. From our analysis of CGTP, leveraging sparsity this takes $\mathcal{O}(\min(s_1 s_2, s_1 s_3, s_2 s_3)) = \mathcal{O}(s_1 s_2 s_3 / \max(s_1, s_2, s_3))$ time. We do this for $\mathcal{O}(L^2)$ points for a total runtime of $\mathcal{O}(s_1 s_2 s_3 L^2 / \max(s_1, s_2, s_3))$ for the interaction.

Finally, we decompose the resulting signal back into tensor harmonic coefficients. First, we can component-wise decompose into scalar SH coefficients. For a fast SH transform, this takes $\mathcal{O}(L^2 \log^2 L)$ time for each component for a total of $\mathcal{O}(s_3 L^2 \log^2 L)$. If we do this for a single $\ell$ it takes $\mathcal{O}(s_3 L^2 \log L)$. Hence, we now have some $B_{m_\ell,m_s}^{\ell}$. To extract the individual components, we use orthogonality of the Clebsch-Gordan coefficients. That is,

$$\sum_{m_\ell,m_s} C_{\ell,m_\ell,s,m_s}^{j,m_j} C_{\ell,m_\ell,s,m_s}^{j,m_j'} = \delta_{m_j,m_j'}.$$

Hence we obtain

$$\mathbf{z}_{m_{j_3}}^{(j_3,\ell_3)} = \sum_{m_\ell,m_s} C_{\ell,m_\ell,s_3,m_s}^{j_3,m_{j_3}} B_{m_\ell,m_s}^{\ell}.$$

Leveraging sparsity of the Clebsch-Gordan coefficients, this takes $\mathcal{O}(s_3 L)$ time to extract a single irrep. To extract all irreps, we just repeat giving $\mathcal{O}(s_3^2 L^2)$ time. Hence, decoding back takes $\mathcal{O}(s_3^2 L^2 + s_3 L^2 \log^2 L)$ time.

Letting $\tilde{s} = \max(s_1, s_2, s_3)$ we the following table which summarizes the runtimes of the various components.

Note if we do not use asymptotically fast version of SH transform but instead a $\mathcal{O}(L^3)$ one, we just change the $L^2 \log^2 L$ terms to $L^3$.

We see that when the $s$'s are fixed constants, the runtimes correspond exactly to those of GTP. Hence our VSTP with $s_1 = s_2 = s_3$ has the same asymptotic runtime as GTP. We also note that at large $L$, MIMO scales with $\tilde{s}$. However, we can

*Table 4.* Asymptotic runtimes of ISTPs assuming a naive grid implementation.

| | SISO | SIMO | MIMO |
|---|---|---|---|
| Encode | $\mathcal{O}((s_1 + s_2)L^2 \log L)$ | $\mathcal{O}((s_1 + s_2)L^2 \log L)$ | $\mathcal{O}((s_1^2 + s_2^2)L^2 + (s_1 + s_2)L^2 \log^2 L)$ |
| Interact | $\mathcal{O}(s_1 s_2 s_3 L^2 / \tilde{s})$ | $\mathcal{O}(s_1 s_2 s_3 L^2 / \tilde{s})$ | $\mathcal{O}(s_1 s_2 s_3 L^2 / \tilde{s})$ |
| Decode | $\mathcal{O}(s_3 L^2 \log L)$ | $\mathcal{O}(s_3^2 L^2 + s_3 L^2 \log^2 L)$ | $\mathcal{O}(s_3^2 L^2 + s_3 L^2 \log^2 L)$ |
| Total | $\mathcal{O}(s_1 s_2 s_3 L^2 / \tilde{s} + \tilde{s} L^2 \log L)$ | $\mathcal{O}((s_1 + s_2)L^2 \log L + s_3^2 L^2 + s_3 L^2 \log^2 L)$ | $\mathcal{O}(\tilde{s}^2 L^2 + \tilde{s} L^2 \log^2 L)$ |

*Table 5.* Asymptotic runtimes of ISTPs assuming asymptotically fast S2FFT grid implementation.

| | SISO | SIMO | MIMO |
|---|---|---|---|
| Encode | $\mathcal{O}((s_1 + s_2)L^2 \log L)$ | $\mathcal{O}((s_1 + s_2)L^2 \log L)$ | $\mathcal{O}((s_1^2 + s_2^2)L^2 + (s_1 + s_2)L^2 \log^2 L)$ |
| Interact | $\mathcal{O}(s_1 s_2 s_3 L^2 / \tilde{s})$ | $\mathcal{O}(s_1 s_2 s_3 L^2 / \tilde{s})$ | $\mathcal{O}(s_1 s_2 s_3 L^2 / \tilde{s})$ |
| Decode | $\mathcal{O}(s_3 L^2 \log L)$ | $\mathcal{O}(s_3^2 L^2 + s_3 L^2 \log^2 L)$ | $\mathcal{O}(s_3^2 L^2 + s_3 L^2 \log^2 L)$ |
| Total | $\mathcal{O}(s_1 s_2 s_3 L^2 / \tilde{s} + \tilde{s} L^2 \log L)$ | $\mathcal{O}((s_1 + s_2)L^2 \log L + s_3^2 L^2 + s_3 L^2 \log^2 L)$ | $\mathcal{O}(\tilde{s}^2 L^2 + \tilde{s} L^2 \log^2 L)$ |

also use $\mathcal{O}(\tilde{s})$ more irreps so in this limit, the additional cost is balanced by performing more tensor products. However, if $L$ is small, we see that runtime scales as $\mathcal{O}(\tilde{s}^2 L^2)$ as we increase $\tilde{s}$. Hence, it still makes sense to minimize the $\tilde{s}$ we use. Note that GTP has $\tilde{s} = 0$ but prevents antisymmetric tensor products, while VSTP has $\tilde{s} = 1$ and the selection rules do not prevent any tensor product paths except the trivial $0 \otimes 0$. Therefore VSTP with $\tilde{s} = 1$ should make the most sense in practice.

### D.2. Asymptotic runtimes in different settings

We compare our results for ISTP and VSTP with a similar table in Xie et al. (2025).

*Table 6.* Asymptotic runtimes of various tensor products for different output settings. The best performing tensor products for each output settings are highlighted in green. In the MIMO setting, the Clebsch-Gordan tensor products are highlighted in red to indicate that they can output irreps with multiplicity $> 1$, unlike the Gaunt tensor products.

| Tensor Product | SISO | SIMO | MIMO |
|---|---|---|---|
| Clebsch-Gordan (Naive) | $\mathcal{O}(L^3)$ | $\mathcal{O}(L^4)$ | $\mathcal{O}(L^6)$ |
| Clebsch-Gordan (Sparse) | $\mathcal{O}(L^2)$ | $\mathcal{O}(L^3)$ | $\mathcal{O}(L^5)$ |
| Gaunt (Original) | $\mathcal{O}(L^2 \log L)$ | $\mathcal{O}(L^3)$ | $\mathcal{O}(L^3)$ |
| Gaunt (Naive Grid) | $\mathcal{O}(L^2 \log L)$ | $\mathcal{O}(L^3)$ | $\mathcal{O}(L^3)$ |
| Gaunt (S2FFT Grid) | $\mathcal{O}(L^2 \log L)$ | $\mathcal{O}(L^2 \log^2 L)$ | $\mathcal{O}(L^2 \log^2 L)$ |
| Vector Signal (Naive Grid) | $\mathcal{O}(L^2 \log L)$ | $\mathcal{O}(L^3)$ | $\mathcal{O}(L^3)$ |
| Vector Signal (S2FFT) | $\mathcal{O}(L^2 \log L)$ | $\mathcal{O}(L^2 \log^2 L)$ | $\mathcal{O}(L^2 \log^2 L)$ |
| ISTP (Naive grid) $(s_1, s_2, s_3)$ | $\mathcal{O}(s_1 s_2 s_3 L^2 / \tilde{s} + \tilde{s} L^2 \log L)$ | $\mathcal{O}((s_1 + s_2)L^2 \log L + s_3^2 L^2 + s_3 L^3)$ | $\mathcal{O}(\tilde{s}^2 L^2 + \tilde{s} L^3)$ |
| ISTP (S2FFT) $(s_1, s_2, s_3)$ | $\mathcal{O}(s_1 s_2 s_3 L^2 / \tilde{s} + \tilde{s} L^2 \log L)$ | $\mathcal{O}((s_1 + s_2)L^2 \log L + s_3^2 L^2 + s_3 L^2 \log^2 L)$ | $\mathcal{O}(\tilde{s}^2 L^2 + \tilde{s} L^2 \log^2 L)$ |

## E. Expressivity

Here, we analyze the expressivity, as defined in Xie et al. (2025), of ISTP. Following their work, we assume we use a tensor product to construct bilinear maps

$$B : (0 \oplus \ldots \oplus L) \times (0 \oplus \ldots \oplus L) \to (0 \oplus \ldots \oplus 2L)$$

by inserting equivariant linear layers before and after the tensor product.

By Schur's lemma, we can only linear maps between irreps of the same type and these maps must be identity. Hence, the total number of inputs and output irreps to our tensor product gives the degrees of freedom for parameterizing the linear layers from $0 \oplus \ldots \oplus L$ and to $0 \oplus \ldots \oplus 2L$. There is an additional 2-fold redundancy in overall scaling so `#Input irreps + #Ouput irreps − 2` gives an upper bound on expressivity.

In the case of ISTP $(s_1, s_2, s_3)$, we note that spherical signals to vector spaces of $\mathbb{R}^{2s+1}$ for a given spin $s$ can be thought of as $2s + 1$ copies of scalar spherical harmonics. In fact, it turns out we need $\mathcal{O}(sL)$ irreps to specify a irrep signal of spin $s$. Hence, the number of input irreps is $\mathcal{O}((s_1 + s_2)L)$ and number of resulting output irreps is $\mathcal{O}(s_3 L)$.

Since VSTP corresponds to a constant choice of $s_1, s_2, s_3$, it ends up having the same asymptotic number of input and output irreps as for GTP.

## F. Proofs

### F.1. Proof of Theorem 4.3

#### F.1.1. DIAGRAMS FOR CLEBSCH-GORDAN COUPLINGS

To assist with the proof, we first introduce a diagrammatic way for working with Clebsch-Gordan couplings. These are inspired by angular momentum diagrams used in quantum mechanics.

We will write a CG coefficient as a point with 3 lines labeled by irrep type and component (orbital and magnetic quantum numbers). Arrows pointing towards the point indicate a bra and arrows pointing away indicate a ket. The number of arrows will determine ordering. Because the CG coefficients are real, flipping all the arrows gives the same coefficient. An example is shown below.

$$C_{\ell,m_\ell,s,m_s}^{j,m_j} = \langle \ell, m_\ell, s, m_s | j, m_j \rangle = \left\{ \vcenter{\hbox{}} \right\} = \langle j, m_j | \ell, m_\ell, s, m_s \rangle = \left\{ \vcenter{\hbox{}} \right\}$$

Note that orientation of the diagram does not matter.

Any variable attached to a leg with a dot will indicate multiplication. Dropping the irrep component (magnetic quantum number) labels means we sum over all components. We will sometimes attach a variable to a leg with no dot to refer to the output. For example

$$\sum_{m_\ell, m_s} C_{\ell,m_\ell,s,m_s}^{j,m_j} \mathbf{x}_{m_\ell}^{(\ell)} \mathbf{y}_{m_s}^{(s)} = \sum_{m_\ell,m_s} \left\{ \vcenter{\hbox{}} \right\} = \left\{ \vcenter{\hbox{}} \right\} = \left\{ \vcenter{\hbox{}} \right\}.$$

In the last diagram, we labeled the output with $\mathbf{z}_{m_j}^{(j)}$ so that it refers to $\mathbf{z}_{m_j}^{(j)} = \sum_{m_\ell,m_s} C_{\ell,m_\ell,s,m_s}^{j,m_j} \mathbf{x}_{m_\ell}^{(\ell)} \mathbf{y}_{m_s}^{(s)}$.

#### F.1.2. USEFUL LEMMAS

**Lemma F.1.**

$$\left\{ \vcenter{\hbox{}} \right\} = (-1)^\ell \sqrt{\frac{2j+1}{2s+1}} \left\{ \vcenter{\hbox{}} \right\}$$

*Proof.* The left represents $\sum_{m_\ell,m_j} C_{\ell,m_\ell,s,m_s}^{j,m_j} \mathbf{Y}_\ell^{m_\ell}(\hat{\mathbf{r}}) \mathbf{x}_{m_j}^{(j)}$. By symmetry relations of the Clebsch-Gordan coefficients, we have

$$C_{\ell,m_\ell,s,m_s}^{j,m_j} = (-1)^{\ell-m_\ell} \sqrt{\frac{2j+1}{2s+1}} C_{j,m_j,\ell,-m_\ell}^{s,m_s}.$$

Substituting, we find

$$
\sum_{m_\ell,m_j} C^{j,m_j}_{\ell,m_\ell,s,m_s}\mathbf{Y}^{m_\ell}_\ell(\hat{\mathbf{r}})\mathbf{x}^{(j)}_{m_j} = \sum_{m_\ell,m_j} (-1)^{\ell-m_\ell}\sqrt{\frac{2j+1}{2s+1}}C^{s,m_s}_{j,m_j,\ell,-m_\ell}\mathbf{Y}^{-m_\ell}_\ell(\hat{\mathbf{r}})\mathbf{x}^{(j)}_{m_j}
$$

$$
= \sum_{m_\ell,m_j} (-1)^\ell\sqrt{\frac{2j+1}{2s+1}}C^{s,m_s}_{j,m_j,\ell,m_\ell}(-1)^{-m_\ell}\mathbf{Y}^{-m_\ell}_\ell(\hat{\mathbf{r}})\mathbf{x}^{(j)}_{m_j}
$$

$$
= \sum_{m_\ell,m_j} (-1)^\ell\sqrt{\frac{2j+1}{2s+1}}C^{s,m_s}_{j,m_j,\ell,m_\ell}\mathbf{Y}^{m_\ell*}_\ell(\hat{\mathbf{r}})\mathbf{x}^{(j)}_{m_j}
$$

where we used the identity $(-1)^{-m_\ell}\mathbf{Y}^{-m_\ell}_\ell(\hat{\mathbf{r}}) = \mathbf{Y}^{m_\ell*}_\ell(\hat{\mathbf{r}})$. The diagram on the right corresponds exactly to the final term, proving the equality. $\qquad\square$

**Lemma F.2.**

$$
\left\{ \begin{array}{c} \mathbf{Y}^*_{\ell_1}(\hat{\mathbf{r}}) \\ \\ \\ \mathbf{Y}^*_{\ell_2}(\hat{\mathbf{r}}) \end{array} \right\} = \sqrt{\frac{(2\ell_1+1)(2\ell_2+1)}{4\pi(2\ell_3+1)}}C^{\ell_3,0}_{\ell_1,0,\ell_2,0}\mathbf{Y}^*_{\ell_3}(\hat{\mathbf{r}})
$$

*Proof.* The diagram represents

$$
\sum_{m_1,m_2} C^{\ell_3,m_3}_{\ell_1,m_1,\ell_2,m_2}\mathbf{Y}^{m_1,*}_{\ell_1}(\hat{\mathbf{r}})\mathbf{Y}^{m_2,*}_{\ell_2}(\hat{\mathbf{r}}).
$$

By the Gaunt formula, we have

$$
\mathbf{Y}^*_{\ell_1}(\hat{\mathbf{r}})\mathbf{Y}^{m_2,*}_{\ell_2}(\hat{\mathbf{r}}) = \sum_{\ell'_3,m'_3}\sqrt{\frac{(2\ell_1+1)(2\ell_2+1)}{4\pi(2\ell_3+1)}}C^{\ell'_3,0}_{\ell_1,0,\ell_2,0}C^{\ell'_3,m'_3}_{\ell_1,m_1,\ell_2,m_2}\mathbf{Y}^{m'_3,*}_{\ell'_3}(\hat{\mathbf{r}}).
$$

Substituting and using an orthogonality relation for the Clebsch-Gordan coefficients, we get

$$
\sum_{m_1,m_2} C^{\ell_3,m_3}_{\ell_1,m_1,\ell_2,m_2}\mathbf{Y}^{m_1,*}_{\ell_1}(\hat{\mathbf{r}})\mathbf{Y}^{m_2,*}_{\ell_2}(\hat{\mathbf{r}})
$$

$$
= \sum_{m_1,m_2} C^{\ell_3,m_3}_{\ell_1,m_1,\ell_2,m_2}\sum_{\ell'_3,m'_3}\sqrt{\frac{(2\ell_1+1)(2\ell_2+1)}{4\pi(2\ell_3+1)}}C^{\ell'_3,0}_{\ell_1,0,\ell_2,0}C^{\ell'_3,m'_3}_{\ell_1,m_1,\ell_2,m_2}\mathbf{Y}^{m'_3,*}_{\ell'_3}(\hat{\mathbf{r}})
$$

$$
= \sum_{\ell'_3,m'_3}\sum_{m_1,m_2} C^{\ell_3,m_3}_{\ell_1,m_1,\ell_2,m_2}C^{\ell'_3,m'_3}_{\ell_1,m_1,\ell_2,m_2}\sqrt{\frac{(2\ell_1+1)(2\ell_2+1)}{4\pi(2\ell_3+1)}}C^{\ell'_3,0}_{\ell_1,0,\ell_2,0}\mathbf{Y}^{m'_3,*}_{\ell'_3}(\hat{\mathbf{r}})
$$

$$
= \sum_{\ell'_3,m'_3}\delta_{\ell_3,\ell'_3}\delta_{m_3,m'_3}\sqrt{\frac{(2\ell_1+1)(2\ell_2+1)}{4\pi(2\ell_3+1)}}C^{\ell'_3,0}_{\ell_1,0,\ell_2,0}\mathbf{Y}^{m'_3,*}_{\ell'_3}(\hat{\mathbf{r}})
$$

$$
= \sqrt{\frac{(2\ell_1+1)(2\ell_2+1)}{4\pi(2\ell_3+1)}}C^{\ell_3,0}_{\ell_1,0,\ell_2,0}\mathbf{Y}^*_{\ell_3}(\hat{\mathbf{r}})
$$

as desired.

Diagrammatically, this computation looks like

$$= \sqrt{\frac{(2\ell_1+1)(2\ell_2+1)}{4\pi(2\ell_3+1)}} C^{\ell_3,0}_{\ell_1,0,\ell_2,0} \mathbf{Y}^*_{\ell_3}(\hat{\mathbf{r}})$$

where the orthogonality relation just corresponds to collapse of the final diagram. $\qquad\square$

### F.1.3. COUPLING OF 4 MOMENTA AND WIGNER 9J

Suppose we have 4 irreps $j_1, \ell_1, j_2, \ell_2$. We can create a tensor product space from these which can be decomposed into irreps. However, there is now ambiguity in how we decompose. We can first couple $j_1, \ell_1$ and decompose into irreps $s_1$, couple $j_2, \ell_2$ and decompose into irreps $s_2$, before finally coupling $s_1, s_2$ and decomposing to final irrep $s_3$. Our basis irreps would then be labeled

$$|(j_1\ell_1)s_1, (j_2\ell_2)s_2, (s_1s_2)s_3\rangle .$$

Conversion from the tensor product space can diagrammatically be drawn as

$$\langle j_1, m_{j_1}, \ell_1, m_{\ell_1}, j_2, m_{j_2}, \ell_2, m_{\ell_2} | (j_1\ell_1)s_1, (j_2\ell_2)s_2, (s_1s_2)s_3, m_{s_3}\rangle$$

Alternatively, we could first couple $j_1, j_2$ to form $j_3$, couple $\ell_1, \ell_2$ to form $\ell_3$, before finally coupling $j_3, \ell_3$ to form $s_3$. This is also a perfectly valid choice of basis which we write as.

$$|(j_1j_2)j_3, (\ell_1\ell_2)\ell_3, (j_3\ell_3)s_3\rangle$$

Similarly, we write conversion from the tensor product space diagrammatically as

$$\langle j_1, m_{j_1}, \ell_1, m_{\ell_1}, j_2, m_{j_2}, \ell_2, m_{\ell_2} | (j_1j_2)j_3, (\ell_1\ell_2)\ell_3, (j_3\ell_3)s_3\rangle$$

The Wigner 9j symbol then tells us how to transform between these basis choices.

**Definition F.3** (Wigner 9j symbol)**.** We denote a Wigner 9j symbol as $3 \times 3$ matrix in curly braces $\begin{Bmatrix} j_1 & \ell_1 & s_1 \\ j_2 & \ell_2 & s_2 \\ j_3 & \ell_3 & s_3 \end{Bmatrix}$. It is defined such that

$$\sqrt{(2s_1+1)(2s_2+1)(2j_3+1)(2\ell_3+1)} \begin{Bmatrix} j_1 & \ell_1 & s_1 \\ j_2 & \ell_2 & s_2 \\ j_3 & \ell_3 & s_3 \end{Bmatrix} = \langle (j_1\ell_1)s_1, (j_2\ell_2)s_2, (s_1s_2)s_3 | (j_1j_2)j_3, (\ell_1\ell_2)\ell_3, (j_3\ell_3)s_3 \rangle .$$

In particular, we can insert the identity to see that

$$|(j_1\ell_1)s_1, (j_2\ell_2)s_2, (s_1s_2)s_3\rangle$$
$$= \sum_{j_3,\ell_3} |(j_1j_2)j_3, (\ell_1\ell_2)\ell_3, (j_3\ell_3)s_3\rangle \langle (j_1j_2)j_3, (\ell_1\ell_2)\ell_3, (j_3\ell_3)s_3 | (j_1\ell_1)s_1, (j_2\ell_2)s_2, (s_1s_2)s_3 \rangle$$
$$= \sum_{j_3,\ell_3} \sqrt{(2s_1+1)(2s_2+1)(2j_3+1)(2\ell_3+1)} \begin{Bmatrix} j_1 & \ell_1 & s_1 \\ j_2 & \ell_2 & s_2 \\ j_3 & \ell_3 & s_3 \end{Bmatrix} |(j_1j_2)j_3, (\ell_1\ell_2)\ell_3, (j_3\ell_3)s_3\rangle$$

We can contract the above with $\langle j_1, m_{j_1}, \ell_1, m_{\ell_1}, j_2, m_{j_2}, \ell_2, m_{\ell_2}|$. Dropping the $m$'s for brevity, this gives the following relation between diagrams

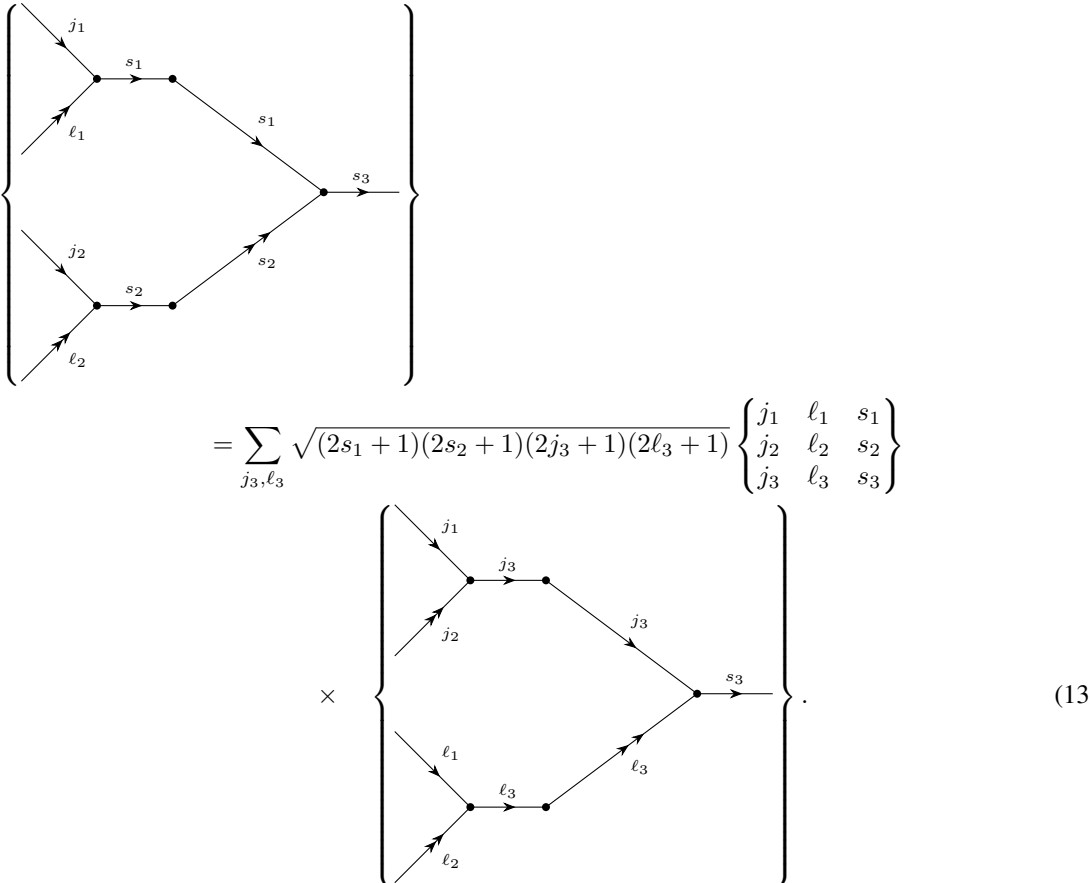

$$= \sum_{j_3,\ell_3} \sqrt{(2s_1+1)(2s_2+1)(2j_3+1)(2\ell_3+1)} \begin{Bmatrix} j_1 & \ell_1 & s_1 \\ j_2 & \ell_2 & s_2 \\ j_3 & \ell_3 & s_3 \end{Bmatrix}$$

$$\times \quad \qquad . \tag{13}$$

F.1.4. PROOF OF THEOREM 4.3

*Proof.* Suppose we have input $\mathbf{x}_m^{(j,\ell)}$ which we couple with TSH $\mathbf{Y}_{j,m}^{\ell,s}$. Then by Definition 4.1, we have

$$\sum_m \mathbf{x}_m^{(j,\ell)} (\mathbf{Y}_{j,m}^{\ell,s}(\hat{\mathbf{r}}))_{m_s} = \sum_m \mathbf{x}_m^{(j,\ell)} \sum_{m_\ell} C_{\ell,m_\ell,s,m_s}^{j,m} \mathbf{Y}_\ell^m(\hat{\mathbf{r}}) = \left\{ \begin{array}{c} \mathbf{x}^{(j,\ell)} \\ \\ \\ \mathbf{Y}_\ell(\hat{\mathbf{r}}) \end{array} \right\}.$$

By Lemma F.1, we can flip arrow directions of this diagram to obtain

$$\sum_m \mathbf{x}_m^{(j,\ell)} \mathbf{Y}_{j,m}^{\ell,s}(\hat{\mathbf{r}}) = (-1)^\ell \sqrt{\frac{2j+1}{2s+1}} \left\{ \begin{array}{c} \mathbf{x}^{(j,\ell)} \\ \\ \\ \mathbf{Y}_\ell^*(\hat{\mathbf{r}}) \end{array} \right\} \tag{14}$$

Suppose we have inputs $\mathbf{x}_1^{(j_1,\ell_1)}$ and $\mathbf{x}_2^{(j_2,\ell_2)}$ which we couple with TSH $\mathbf{Y}_{j_1,m_1}^{\ell_1,s_1}$ and $\mathbf{Y}_{j_2,m_2}^{\ell_2,s_2}$ respectively. We can diagrammatically write

$$\left( \sum_{m_1} \mathbf{x}_{1,m_1}^{(j_1,\ell_1)} \mathbf{Y}_{j_1,m_1}^{\ell_1,s_1}(\hat{\mathbf{r}}) \otimes \sum_{m_2} \mathbf{x}_{2,m_2}^{(j_2,\ell_2)} \mathbf{Y}_{j_2,m_2}^{\ell_2,s_2}(\hat{\mathbf{r}}) \right)^{s_3} = \left\{ \begin{array}{c} \sum_{m_1} \mathbf{x}_{1,m_1}^{(j_1,\ell_1)} \mathbf{Y}_{j_1,m_1}^{\ell_1,s_1}(\hat{\mathbf{r}}) \\ \\ \\ \sum_{m_2} \mathbf{x}_{2,m_2}^{(j_2,\ell_2)} \mathbf{Y}_{j_2,m_2}^{\ell_2,s_2}(\hat{\mathbf{r}}) \end{array} \right\}$$

We can then expand the summations using equation 14. This gives the following diagram.

$$\left( \sum_{m_1} \mathbf{x}_{1,m_1}^{(j_1,\ell_1)} \mathbf{Y}_{j_1,m_1}^{\ell_1,s_1}(\hat{\mathbf{r}}) \otimes \sum_{m_2} \mathbf{x}_{2,m_2}^{(j_2,\ell_2)} \mathbf{Y}_{j_2,m_2}^{\ell_2,s_2}(\hat{\mathbf{r}}) \right)^{s_3}$$

$$= (-1)^{\ell_1} \sqrt{\frac{2j_1+1}{2s_1+1}} (-1)^{\ell_2} \sqrt{\frac{2j_2+1}{2s_2+1}} \left\{ \begin{array}{c} \mathbf{x}_1^{(j_1,\ell_1)} \\ \\ \mathbf{Y}_{\ell_1}^*(\hat{\mathbf{r}}) \\ \mathbf{x}_2^{(j_2,\ell_2)} \\ \\ \mathbf{Y}_{\ell_2}^*(\hat{\mathbf{r}}) \end{array} \right\}.$$

It is now apparent that we have the coupling structure of 4 irreps. In particular, we first couple our input and spherical harmonic pairs $(j_i, \ell_i)$ to get our irrep signals $(s_i)$ before coupling the signals $(s_1, s_2)$. Alternatively we could have first

coupled the input irreps and the spherical harmonics. Using Equation 13, we can rewrite using this alternative coupling as

$$
\left( \sum_{m_1} \mathbf{x}_{1,m_1}^{(j_1,\ell_1)} \mathbf{Y}_{j_1,m_1}^{\ell_1,s_1}(\hat{\mathbf{r}}) \otimes \sum_{m_2} \mathbf{x}_{2,m_2}^{(j_2,\ell_2)} \mathbf{Y}_{j_2,m_2}^{\ell_2,s_2}(\hat{\mathbf{r}}) \right)^{s_3}
$$

$$
= (-1)^{\ell_1} \sqrt{\frac{2j_1+1}{2s_1+1}} (-1)^{\ell_2} \sqrt{\frac{2j_2+1}{2s_2+1}} \sum_{j_3,\ell_3} \sqrt{(2s_1+1)(2s_2+1)(2j_3+1)(2\ell_3+1)} \begin{Bmatrix} j_1 & \ell_1 & s_1 \\ j_2 & \ell_2 & s_2 \\ j_3 & \ell_3 & s_3 \end{Bmatrix}
$$

$$.$$

Applying Lemma F.2, the lower branch becomes $\sqrt{\frac{(2\ell_1+1)(2\ell_2+1)}{4\pi(2\ell_3+1)}} C_{\ell_1,0,\ell_2,0}^{\ell_3,0} \mathbf{Y}_{\ell_3}^*(\hat{\mathbf{r}})$.

$$\left( \sum_{m_1} \mathbf{x}_{1,m_1}^{(j_1,\ell_1)} \mathbf{Y}_{j_1,m_1}^{\ell_1,s_1}(\hat{\mathbf{r}}) \otimes \sum_{m_2} \mathbf{x}_{2,m_2}^{(j_2,\ell_2)} \mathbf{Y}_{j_2,m_2}^{\ell_2,s_2}(\hat{\mathbf{r}}) \right)^{s_3}$$

$$= (-1)^{\ell_1} \sqrt{\frac{2j_1+1}{2s_1+1}} (-1)^{\ell_2} \sqrt{\frac{2j_2+1}{2s_2+1}} \sum_{j_3,\ell_3} \sqrt{(2s_1+1)(2s_2+1)(2j_3+1)(2\ell_3+1)} \begin{Bmatrix} j_1 & \ell_1 & s_1 \\ j_2 & \ell_2 & s_2 \\ j_3 & \ell_3 & s_3 \end{Bmatrix}$$

$$= (-1)^{\ell_1} \sqrt{\frac{2j_1+1}{2s_1+1}} (-1)^{\ell_2} \sqrt{\frac{2j_2+1}{2s_2+1}} \sum_{j_3,\ell_3} \sqrt{(2s_1+1)(2s_2+1)(2j_3+1)(2\ell_3+1)} \begin{Bmatrix} j_1 & \ell_1 & s_1 \\ j_2 & \ell_2 & s_2 \\ j_3 & \ell_3 & s_3 \end{Bmatrix}$$

$$\sqrt{\frac{(2\ell_1+1)(2\ell_2+1)}{4\pi(2\ell_3+1)}} C_{\ell_1,0,\ell_2,0}^{\ell_3,0} (-1)^{\ell_3} \sqrt{\frac{2s_3+1}{2j_3+1}} (-1)^{\ell_3} \sqrt{\frac{2j_3+1}{2s_3+1}}$$

where we pulled out constant factors and inserted $1 = (-1)^{\ell_3} \sqrt{\frac{2s_3+1}{2j_3+1}} (-1)^{\ell_3} \sqrt{\frac{2j_3+1}{2s_3+1}}$. Comparing the diagram with Equation 14, we see we have exactly a TSH coupling. This gives,

$$\left( \sum_{m_1} \mathbf{x}_{1,m_1}^{(j_1,\ell_1)} \mathbf{Y}_{j_1,m_1}^{\ell_1,s_1}(\hat{\mathbf{r}}) \otimes \sum_{m_2} \mathbf{x}_{2,m_2}^{(j_2,\ell_2)} \mathbf{Y}_{j_2,m_2}^{\ell_2,s_2}(\hat{\mathbf{r}}) \right)^{s_3}$$

$$= (-1)^{\ell_1} \sqrt{\frac{2j_1+1}{2s_1+1}} (-1)^{\ell_2} \sqrt{\frac{2j_2+1}{2s_2+1}} \sum_{j_3,\ell_3} \sqrt{(2s_1+1)(2s_2+1)(2j_3+1)(2\ell_3+1)} \begin{Bmatrix} j_1 & \ell_1 & s_1 \\ j_2 & \ell_2 & s_2 \\ j_3 & \ell_3 & s_3 \end{Bmatrix}$$

$$\sqrt{\frac{(2\ell_1+1)(2\ell_2+1)}{4\pi(2\ell_3+1)}} C_{\ell_1,0,\ell_2,0}^{\ell_3,0} (-1)^{\ell_3} \sqrt{\frac{2s_3+1}{2j_3+1}} \sum_{m_3}$$  $$\mathbf{Y}_{j_3,m_3}^{\ell_3,s_3}(\hat{\mathbf{r}}).$$

We simplify the constants

$$
\left(\sum_{m_1}\mathbf{x}_{1,m_1}^{(j_1,\ell_1)}\mathbf{Y}_{j_1,m_1}^{\ell_1,s_1}(\hat{\mathbf{r}})\otimes\sum_{m_2}\mathbf{x}_{2,m_2}^{(j_2,\ell_2)}\mathbf{Y}_{j_2,m_2}^{\ell_2,s_2}(\hat{\mathbf{r}})\right)^{s_3}
$$

$$
=\sqrt{\frac{(2j_1+1)(2j_2+1)(2\ell_1+1)(2\ell_2+1)(2s_3+1)}{4\pi}}
$$

$$
\sum_{j_3,\ell_3}\begin{Bmatrix}j_1 & \ell_1 & s_1\\ j_2 & \ell_2 & s_2\\ j_3 & \ell_3 & s_3\end{Bmatrix}(-1)^{\ell_1+\ell_2+\ell_3}C_{\ell_1,0,\ell_2,0}^{\ell_3,0}\sum_{m_3}\left\{\begin{array}{c}\mathbf{x}_1^{(j_1,\ell_1)}\\ \\ \\ \\ \mathbf{x}_2^{(j_2,\ell_2)}\end{array}\right.\raisebox{0pt}{\includegraphics}\left.\begin{array}{c}\\ j_3,m_3\\ \\ \end{array}\right\}\mathbf{Y}_{j_3,m_3}^{\ell_3,s_3}(\hat{\mathbf{r}})
$$

$$
=\sqrt{\frac{(2j_1+1)(2j_2+1)(2\ell_1+1)(2\ell_2+1)(2s_3+1)}{4\pi}}
$$

$$
\sum_{j_3,\ell_3}\begin{Bmatrix}j_1 & \ell_1 & s_1\\ j_2 & \ell_2 & s_2\\ j_3 & \ell_3 & s_3\end{Bmatrix}C_{\ell_1,0,\ell_2,0}^{\ell_3,0}\sum_{m_3}\sum_{m_1,m_2}C_{j_1,m_1,j_2,m_2}^{j_3,m_3}\mathbf{x}_{1,m_1}^{(j_1,\ell_1)}\mathbf{x}_{2,m_2}^{(j_2,\ell_2)}\mathbf{Y}_{j_3,m_3}^{\ell_3,s_3}(\hat{\mathbf{r}})
$$

$$
=\sqrt{\frac{(2j_1+1)(2j_2+1)(2\ell_1+1)(2\ell_2+1)(2s_3+1)}{4\pi}}
$$

$$
\sum_{m_1,m_2}\mathbf{x}_{1,m_1}^{(j_1,\ell_1)}\mathbf{x}_{2,m_2}^{(j_2,\ell_2)}\sum_{j_3,\ell_3,m_3}\begin{Bmatrix}j_1 & \ell_1 & s_1\\ j_2 & \ell_2 & s_2\\ j_3 & \ell_3 & s_3\end{Bmatrix}C_{\ell_1,0,\ell_2,0}^{\ell_3,0}C_{j_1,m_1,j_2,m_2}^{j_3,m_3}\mathbf{Y}_{j_3,m_3}^{\ell_3,s_3}(\hat{\mathbf{r}})
$$

where we noted that $C_{\ell_1,0,\ell_2,0}^{\ell_3,0}$ enforces $\ell_1+\ell_2+\ell_3$ must be even so $(-1)^{\ell_1+\ell_2+\ell_3}$ vanishes. Finally by bilinearity, we can also pull out the summations and $\mathbf{x}_{1,m_1}^{(j_1,\ell_1)},\mathbf{x}_{2,m_2}^{(j_2,\ell_2)}$ in the first term. Hence, we obtain

$$
\left(\sum_{m_1}\mathbf{x}_{1,m_1}^{(j_1,\ell_1)}\mathbf{Y}_{j_1,m_1}^{\ell_1,s_1}(\hat{\mathbf{r}})\otimes\sum_{m_2}\mathbf{x}_{2,m_2}^{(j_2,\ell_2)}\mathbf{Y}_{j_2,m_2}^{\ell_2,s_2}(\hat{\mathbf{r}})\right)^{s_3}
$$

$$
=\sum_{m_1,m_2}\mathbf{x}_{1,m_1}^{(j_1,\ell_1)}\mathbf{x}_{2,m_2}^{(j_2,\ell_2)}\left(\mathbf{Y}_{j_1,m_1}^{\ell_1,s_1}(\hat{\mathbf{r}})\otimes\mathbf{Y}_{j_2,m_2}^{\ell_2,s_2}(\hat{\mathbf{r}})\right)^{s_3}
$$

$$
=\sqrt{\frac{(2j_1+1)(2j_2+1)(2\ell_1+1)(2\ell_2+1)(2s_3+1)}{4\pi}}
$$

$$
\sum_{m_1,m_2}\mathbf{x}_{1,m_1}^{(j_1,\ell_1)}\mathbf{x}_{2,m_2}^{(j_2,\ell_2)}\sum_{j_3,\ell_3,m_3}\begin{Bmatrix}j_1 & \ell_1 & s_1\\ j_2 & \ell_2 & s_2\\ j_3 & \ell_3 & s_3\end{Bmatrix}C_{\ell_1,0,\ell_2,0}^{\ell_3,0}C_{j_1,m_1,j_2,m_2}^{j_3,m_3}\mathbf{Y}_{j_3,m_3}^{\ell_3,s_3}(\hat{\mathbf{r}}).
$$

Since this must hold for all values of $\mathbf{x}_{1,m_1}^{(j_1,\ell_1)},\mathbf{x}_{2,m_2}^{(j_2,\ell_2)}$, this implies

$$
\left(\mathbf{Y}_{j_1,m_1}^{\ell_1,s_1}(\hat{\mathbf{r}})\otimes\mathbf{Y}_{j_2,m_2}^{\ell_2,s_2}(\hat{\mathbf{r}})\right)^{s_3}
$$

$$
=\sqrt{\frac{(2j_1+1)(2j_2+1)(2\ell_1+1)(2\ell_2+1)(2s_3+1)}{4\pi}}\sum_{j_3,\ell_3,m_3}\begin{Bmatrix}j_1 & \ell_1 & s_1\\ j_2 & \ell_2 & s_2\\ j_3 & \ell_3 & s_3\end{Bmatrix}C_{\ell_1,0,\ell_2,0}^{\ell_3,0}C_{j_1,m_1,j_2,m_2}^{j_3,m_3}\mathbf{Y}_{j_3,m_3}^{\ell_3,s_3}(\hat{\mathbf{r}}).
$$

$\square$

**F.2. Proof of Theorem 5.1**

*Proof.* Theorem 4.3 tells us that

$$(\mathbf{Y}_{\ell_1,1}^{j_1,m_1} \otimes \mathbf{Y}_{\ell_2,1}^{j_2,m_2})_1$$

$$= \sqrt{\frac{(2j_1+1)(2j_2+1)(2\ell_1+1)(2\ell_2+1)3}{4\pi}} \sum_{j_3,\ell_3,m_3} \begin{Bmatrix} j_1 & \ell_1 & 1 \\ j_2 & \ell_2 & 1 \\ j_3 & \ell_3 & 1 \end{Bmatrix} C_{\ell_1,0,\ell_2,0}^{\ell_3,0} C_{j_1,m_1,j_2,m_2}^{j_3,m_3} \mathbf{Y}_{\ell_3,1}^{j_3,m_3}.$$

We first work through the only if direction.

Rules 1-3 follow from the fact that the triangle condition must be satisfied in the rows and columns of the Wigner 9j symbol. Rule 4 follows from the even selection rule of the $C_{\ell_1,0,\ell_2,0}^{\ell_3,0}$ coefficient.

For selection rule 5, we use the symmetry properties of the Wigner 9j symbols. Suppose there exists a permutation $a, b, c$ such that $j_a = \ell_a$ and $(j_b, \ell_b) = (j_c, \ell_c)$. Suppose we swap rows $b, c$, this is an odd permutation so the symmetries of the 9j symbol means we pick up a phase factor $(-1)^S$ where $S = \sum_{i=1}^{3}(j_i + \ell_i + 1)$. Note that $S$ is odd because each $j_b + \ell_b = j_c + \ell_c$ and $j_a = \ell_a$. Hence our phase factor is $-1$. But swapping $b, c$ does not change the 9j symbol since $j_b = j_c$. Hence by symmetry the 9j symbol must vanish, giving us selection rule 5.

To check the if direction, we just need to check that the Wigner 9j and the $C_{\ell_1,0,\ell_2,0}^{\ell,0}$ are nonzero when the rules are satisfied. First, it is known that $C_{\ell_1,0,\ell_2,0}^{\ell,0}$ is nonzero if and only if it satisfies rule 4 (Raynal, 1978). For the Wigner 9j symbols, we use the explicit formulas documented in Varshalovich et al. (1988). These are listed in tables where the Wigner 9j takes the following form

$$\begin{Bmatrix} a+\lambda & a & 1 \\ b+\mu & b & 1 \\ c+\nu & c & 1 \end{Bmatrix}.$$

The modified versions of rules 1-3 and rule 5 with this change of variables is as follows

1. $\{a+\lambda, a, 1\} = \{b+\mu, b, 1\} = \{c+\nu, c, 1\} = 1$

2. $\{a+\lambda, b+\mu, c+\nu\} = 1$

3. $\{a, b, c\} = 1$

5. (a) We do not have $\lambda = 0$ and $b = c$ and $\mu = \nu$

   (b) We do not have $\mu = 0$ and $a = c$ and $\lambda = \nu$

   (c) We do not have $\nu = 0$ and $a = b$ and $\lambda = \mu$

We now work through the formula tables by casework on $\lambda, \mu, \nu$. For brevity, we introduce $s = a + b + c$. All formulas consist of products linear or factorial terms in the numerator and denominator. We simply inspect the linear terms in the numerator and check which terms potentially can be 0. We also check the denominator factorials which can potentially become infinite for negative integers (assuming extension to gamma functions). We then check which rules the problematic terms would break.

We list this casework as a series of 3 tables separated by values of $\nu$. First two columns indicate the value of $\lambda, \mu$ considered. Then we list the formula from Varshalovich et al. (1988). Then we list the problematic terms. Finally, we list the rules the problem terms would break, respectively.

For $\nu = 1$:

| $\lambda$ | $\mu$ | Formula | 0 terms | Rules broken |
|---|---|---|---|---|
| 1 | 1 | $-\left[\dfrac{(s+4)!(s-2c+1)(s-2b+1)(s-2a+1)(2a)!(2b)!(2c)!}{3(s+1)!(2a+3)!(2b+3)!(2c+3)!}\right]^{1/2}$ | $(s-2c+1), (s-2b+1), (s-2a+1)$ | 3, 3, 3 |
| 1 | 0 | $(c-a)\left[\dfrac{2(s+3)!(s-2b+2)(2a)!(2b-1)!(2c)!}{3(s+1)!(s-2b)!(2a+3)!(2b+2)!(2c+3)!}\right]^{1/2}$ | $(c-a), (s-2b+2), \frac{1}{(s-2b)!}$ | 5b, 3, 3 |
| 1 | -1 | $-\left[\dfrac{(s+2)(s-2c)(s-2b+3)!(s-2a)(2a)!(2b-2)!(2c)!}{3(s-2b)!(2a+3)!(2b+1)!(2c+3)!}\right]^{1/2}$ | $(s-2c), (s-2a), \frac{1}{(s-2b)!}$ | 2, 2, 3 |
| 0 | 1 | $(b-c)\left[\dfrac{2(s+3)!(s-2a+2)(2a-1)!(2b)!(2c)!}{3(s+1)!(s-2a)!(2a+2)!(2b+3)!(2c+3)!}\right]^{1/2}$ | $(b-c), (s-2a+2), \frac{1}{(s-2a)!}$ | 5a, 3, 2 |
| 0 | 0 | $2(c+1)\left[\dfrac{(s+2)(s-2c)(s-2b+1)(s-2a+1)(2a-1)!(2b-1)!(2c)!}{3(2a+2)!(2b+2)!(2c+3)!}\right]^{1/2}$ | $(s-2c), (s-2b+1), (s-2a+1)$ | 2, 3, 3 |
| 0 | -1 | $-(c+b+1)\left[\dfrac{2(s-2c)!(s-2b+2)!(2a-1)!(2b-2)!(2c)!}{3(s-2c-2)!(s-2b)!(2a+2)!(2b+1)!(2c+3)!}\right]^{1/2}$ | $\frac{1}{(s-2c-2)!}, \frac{1}{(s-2b)!}$ | 2, 3 |
| -1 | 1 | $-\left[\dfrac{(s+2)(s-2c)(s-2b)(s-2a+3)!(2a-2)!(2b)!(2c)!}{3(s-2a)!(2a+1)!(2b+3)!(2c+3)!}\right]^{1/2}$ | $(s-2c), (s-2b), \frac{1}{(s-2a)!}$ | 2, 2, 3 |
| -1 | 0 | $(a+c+1)\left[\dfrac{2(s-2c)!(s-2a+2)!(2a-2)!(2b-1)!(2c)!}{3(s-2c-2)!(s-2a)!(2a+1)!(2b+2)!(2c+3)!}\right]^{1/2}$ | $\frac{1}{(s-2c-2)!}, \frac{1}{(s-2a)!}$ | 2, 3 |
| -1 | -1 | $-\left[\dfrac{(s+1)(s-2c)!(s-2b+1)(s-2a+1)(2a-2)!(2b-2)!(2c)!}{3(s-2c-3)!(2a+1)!(2b+1)!(2c+3)!}\right]^{1/2}$ | $(s-2b+1), (s-2a+1), \frac{1}{(s-2c-3)!}$ | 3, 3, 2 |

For $\nu = 0$:

| $\lambda$ | $\mu$ | Formula | 0 terms | Rules broken |
|---|---|---|---|---|
| 1 | 1 | $(a-b)\left[\dfrac{2(s+3)!(s-2c+2)(2a)!(2b)!(2c-1)!}{3(s+1)!(s-2c)!(2a+3)!(2b+3)!(2c+2)!}\right]^{1/2}$ | $(a-b), (s-2c+2), \frac{1}{(s-2c)!}$ | 5c, 3, 3 |
| 1 | 0 | $2(a+1)\left[\dfrac{(s+2)(s-2c+1)(s-2b+1)(s-2a)(2a)!(2b-1)!(2c-1)!}{3(2a+3)!(2b+2)!(2c+2)!}\right]^{1/2}$ | $(s-2c+1), (s-2b+1), (s-2a)$ | 3, 3, 2 |
| 1 | -1 | $(a+b+1)\left[\dfrac{2(s-2b+2)!(s-2a)(2a)!(2b-2)!(2c-1)!}{3(s-2b)!(s-2a-2)!(2a+3)!(2b+1)!(2c+2)!}\right]^{1/2}$ | $\frac{1}{(s-2b)!}, \frac{1}{(s-2a-2)!}$ | 3, 2 |
| 0 | 1 | $2(b+1)\left[\dfrac{(s+2)(s-2c+1)(s-2b)(s-2a+1)(2a-1)!(2b)!(2c-1)!}{3(2a+2)!(2b+3)!(2c+2)!}\right]^{1/2}$ | $(s-2c+1), (s-2b), (s-2a+1)$ | 3, 2, 3 |
| 0 | 0 | $0$ | Entire expression | 5abc |
| 0 | -1 | $2b\left[\dfrac{(s+1)(s-2c)(s-2b+1)(s-2a)(2a-1)!(2b-2)!(2c-1)!}{3(2a+2)!(2b+1)!(2c+2)!}\right]^{1/2}$ | $(s-2c), (s-2b+1), (s-2a)$ | 2, 3, 2 |
| -1 | 1 | $-(a+b+1)\left[\dfrac{2(s-2b)!(s-2a+2)!(2a-2)!(2b)!(2c-1)!}{3(s-2b-2)!(s-2a)!(2a+1)!(2b+3)!(2c+2)!}\right]^{1/2}$ | $\frac{1}{(s-2b-2)!}, \frac{1}{(s-2a)!}$ | 2, 3 |
| -1 | 0 | $2a\left[\dfrac{(s+1)(s-2c)(s-2b)(s-2a+1)(2a-2)!(2b-1)!(2c-1)!}{3(2a+1)!(2b+2)!(2c+2)!}\right]^{1/2}$ | $(s-2c), (s-2b), (s-2a+1)$ | 2, 2, 3 |
| -1 | -1 | $(b-a)\left[\dfrac{2(s+1)!(s-2c)!(2a-2)!(2b-2)!(2c-1)!}{3(s-1)!(s-2c-2)!(2a+1)!(2b+1)!(2c+2)!}\right]^{1/2}$ | $(b-a), \frac{1}{(s-1)!}, \frac{1}{(s-2c-2)!}$ | 5c, 1, 2 |

For $\nu = -1$:

| $\lambda$ | $\mu$ | Formula | 0 terms | Rules broken |
|---|---|---|---|---|
| 1 | 1 | $-\left[\dfrac{(s+2)(s-2c+3)!(s-2b)(s-2a)(2a)!(2b)!(2c-2)!}{3(s-2c)!(2a+3)!(2b+3)!(2c+1)!}\right]^{1/2}$ | $(s-2b), (s-2a), \frac{1}{(s-2c)!}$ | 2, 2, 3 |
| 1 | 0 | $-(a+c+1)\left[\dfrac{2(s-2c+2)!(s-2a)(2a)!(2b-1)!(2c-2)!}{3(s-2c)!(s-2a-2)!(2a+3)!(2b+2)!(2c+1)!}\right]^{1/2}$ | $\frac{1}{(s-2c)!}, \frac{1}{(s-2a-2)!}$ | 3, 2 |
| 1 | -1 | $-\left[\dfrac{(s+1)(s-2c+1)(s-2b+1)(s-2a)!(2a)!(2b-2)!(2c-2)!}{3(s-2a-3)!(2a+3)!(2b+1)!(2c+1)!}\right]^{1/2}$ | $(s-2c+1), (s-2b+1), \frac{1}{(s-2a-3)!}$ | 3, 3, 2 |
| 0 | 1 | $(b+c+1)\left[\dfrac{2(s-2c+2)!(s-2b)!(2a-1)!(2b)!(2c-2)!}{3(s-2c)!(s-2b-2)!(2a+2)!(2b+3)!(2c+1)!}\right]^{1/2}$ | $\frac{1}{(s-2c)!}, \frac{1}{(s-2b-2)!}$ | 3, 2 |
| 0 | 0 | $2c\left[\dfrac{(s+1)(s-2c+1)(s-2b)(s-2a)(2a-1)!(2b-1)!(2c-2)!}{(2a+2)!(2b+2)!(2c+1)!}\right]^{1/2}$ | $(s-2c+1), (s-2b), (s-2a)$ | 3, 2, 2 |
| 0 | -1 | $(c-b)\left[\dfrac{2(s+1)!(s-2a)!(2a-1)!(2b-2)!(2c-2)!}{3(s-1)!(s-2a-2)!(2a+2)!(2b+1)!(2c+1)!}\right]^{1/2}$ | $(c-b), \frac{1}{(s-1)!}, \frac{1}{(s-2a-2)!}$ | 5a, 1, 2 |
| -1 | 1 | $-\left[\dfrac{(s+1)(s-2c+1)(s-2b)(s-2a+1)(2a-2)!(2b)!(2c-2)!}{3(s-2b-3)!(2a+1)!(2b+3)!(2c+1)!}\right]^{1/2}$ | $(s-2c+1), (s-2a+1), \frac{1}{(s-2b-3)!}$ | 3, 3, 2 |
| -1 | 0 | $(a-c)\left[\dfrac{2(s+1)!(s-2b)!(2a-2)!(2b-1)!(2c-2)!}{3(s-1)!(s-2b-2)!(2a+1)!(2b+2)!(2c+1)!}\right]^{1/2}$ | $(a-c), \frac{1}{(s-1)!}, \frac{1}{(s-2b-2)!}$ | 5b, 1, 2 |
| -1 | -1 | $-\left[\dfrac{(s+1)!(s-2c)(s-2b)(s-2a)(2a-2)!(2b-2)!(2c-2)!}{3(s-2)!(2a+1)!(2b+1)!(2c+1)!}\right]^{1/2}$ | $(s-2c), (s-2b), (s-2a), \frac{1}{(s-2)!}$ | 2, 2, 2, 1 |

From this careful casework, we see the only way for the Wigner 9js to be 0 are if they break the selection rules 1-3 or 5. Hence, we conclude it is nonzero if and only if selection rules 1-3 and 5 are satisfied.

Combining, with the fact that $C^{\ell_3,0}_{\ell_1,0,\ell_2,0}$ is nonzero if and only if rule 4 is satisfied, we conclude the VSTP interaction is nonzero if and only if all the selection rules 1-5 are satisfied.

$\square$

## F.3. Proof of Theorem 5.2

*Proof.* We already satisfy condition 2. Without loss of generality, assume $j_1 \leq j_2 \leq j_3$

*Case 1:* Suppose $j_1, j_2, j_3$ are distinct. Condition 5 is already satisfied since the $j$'s are unique. Since the $j$'s are distinct integers, we have $j_1 + 1 \leq j_2 \leq j_3 - 1$. If $j_1 + j_2 + j_3$ is even, we can set $\ell_i = j_i$ so conditions $1, 3, 4$ are clearly satisfied. If $j_1 + j_2 + j_3$ is odd, we can set $\ell_1 = j_1, \ell_2 = j_2, \ell_3 = j - 3$. By construction $1, 4$ are satisfied. For 3, we have

$$\ell_3 < j_3 \leq j_1 + j_2 = \ell_1 + \ell_2$$

$$\ell_2 = j_2 \leq j_3 - 1 = \ell_3 \leq \ell_3 + \ell_1$$

$$\ell_1 = j_1 \leq j_3 - 1 = \ell_3 \leq \ell_3 + \ell_2.$$

Hence we can always choose $\ell_1, \ell_2, \ell_3$ which satisfy the selection rules.

*Case 2:* Suppose two of $j_1, j_2, j_3$ are equal. Then we have two subcases.

*Subcase 1:* $j_1 = j_2 \leq j_3 - 1$. Note that $j_3 - 1 \geq 0$ so $j_3 \geq 1$. But $1 \leq j_3 \leq j_1 + j_2 = 2j_1$ so $j_1 \geq 1$ since $j_1$ is an integer.

If $j_1 + j_2 + j_3$ is even, we can set $\ell_1 = j_1, \ell_2 = j_2 + 1$, and $\ell_3 = j_3 - 1$. By construction we satisfy $1, 4$. Since $\ell_1 \neq \ell_2$ and $j_1 = j_2 \neq j_3$ we will satisfy 5. For 3 we find that

$$\ell_1 = j_1 = j_2 < j_2 + 1 = \ell_2 \leq \ell_2 + \ell_3$$

$$\ell_2 = j_2 + 1 \leq j_3 = \ell_3 + 1 \leq \ell_1 + \ell_3$$

$$\ell_3 = j_3 - 1 < j_1 + j_2 = \ell_1 + \ell_2 - 1 < \ell_1 + \ell_2.$$

Hence we satisfy all the selection rules.

Suppose $j_1 + j_2 + j_3$ is odd. Then we can set $\ell_1 = j_1, \ell_2 = j_2 + 1$, and $\ell_3 = j_3$. By construction we satisfy $1, 4$. Since $\ell_1 \neq \ell_2$ and $j_1 = j_2 \neq j_3$ we will satisfy 5. For 3 we find that

$$\ell_1 = j_1 = j_2 < j_2 + 1 = \ell_2 \leq \ell_2 + \ell_3$$

$$\ell_2 = j_2 + 1 \le j_3 = \ell_3 \le \ell_1 + \ell_3$$

$$\ell_3 = j_3 \le j_1 + j_2 = \ell_1 + \ell_2 - 1 < \ell_1 + \ell_2.$$

Hence we satisfy all the selection rules.

*Subcase 2:* $j_1 + 1 \le j_2 = j_3$. If $j_1 + j_2 + j_3$ is even, we can set $\ell_1 = j_1 + 1$, $\ell_2 = j_2$, and $\ell_3 = j_3 - 1$. By construction we satisfy $1, 4$. Since $\ell_2 \ne \ell_3$ and $j_2 = j_3 \ne j_1$, we satisfy 5. For 3 we find

$$\ell_1 = j_1 + 1 \le j_2 = \ell_2 \le \ell_2 + \ell_3$$

$$\ell_2 = j_2 \le j_1 + j_3 = (j_1 + 1) + (j_3 - 1) = \ell_1 + \ell_3$$

$$\ell_3 = j_3 - 1 \le j_1 + j_2 - 1 < j_1 + 1 + j_2 = \ell_1 + \ell_2.$$

Hence we satisfy all the selection rules.

If $j_1 + j_2 + j_3$ is odd, we can set $\ell_1 = j_1 + 1$, $\ell_2 = j_2$, and $\ell_3 = j_3$. By construction we satisfy $1, 4$. Since $j_1 \ne \ell_1$ and $j_2 = j_3 \ne j_1$, we satisfy 5. For 3 we find

$$\ell_1 = j_1 + 1 \le j_3 = \ell_3 \le \ell_2 + \ell_3$$

$$\ell_2 = j_2 \le j_1 + j_3 < (j_1 + 1) + j_3 = \ell_1 + \ell_3$$

$$\ell_3 = j_3 \le j_1 + j_2 < (j_1 + 1) + j_2 = \ell_1 + \ell_2.$$

Hence we satisfy all the selection rules.

*Case 3:* Suppose $j_1 = j_2 = j_3 = j$. If $j > 0$ and is even, then $j \ge 2$. We can pick $\ell_1 = j - 1$, $\ell_2 = j$, $\ell_3 = j + 1$. By construction we satisfy $1, 4$. Since the $\ell$'s are distinct we also satisfy 5. For 3 we check that

$$\ell_1 = j - 1 < j + j + 1 = \ell_2 + \ell_3$$

$$\ell_2 = j < j - 1 + j + 1 = \ell_1 + \ell_3$$

$$\ell_3 = j + 1 = j - 1 + 2 \le j - 1 + j = \ell_1 + \ell_2.$$

Hence we satisfy all the selection rules.

If $j$ is odd then $j \ge 1$. We can pick $\ell_1 = j - 1$, $\ell_2 = j$, $\ell_3 = j$. By construction we satisfy $1, 4$. Since $\ell_1 \ne j$ and $\ell_2 = \ell_3 \ne \ell_1$ we also satisfy 5. For 3 we check that

$$\ell_1 = j - 1 < j + j = \ell_2 + \ell_3$$

$$\ell_2 = j \le j - 1 + j = \ell_1 + \ell_3$$

$$\ell_3 = j \le j - 1 + j = \ell_1 + \ell_2.$$

Hence we satisfy all the selection rules.

The only case which fails is $j_1 = j_2 = j_3 = 0$ in which case selection rule 1 forces $\ell_1 = \ell_2 = \ell_3 = 1$ which breaks rule 4. However, this case just correspond to multiplication of scalars which is trivial. □

## G. Details of Message-Passing Network

In Algorithm 1, we create learnable (ie, parametrized) variants of the purely functional tensor products. For the Clebsch-Gordan tensor product $\otimes_{\text{CG}}$, we simply add a linear layer to its output. For the Gaunt tensor product $\otimes_{\text{GTP}}$, we create multiple channels, perform the tensor product channel-wise and then concatenate all irreps. This allows the output to have irreps of multiplicity $> 1$, even with the Gaunt tensor product. We set the number of channels $C$ as 4 in all experiments with the Gaunt tensor product.

In Algorithm 2, we use these learnable tensor products in a simple message-passing network, very similar to NequIP (Batzner et al., 2022).

---
**Algorithm 1** LEARNABLETENSORPRODUCT
---

**Require:** Tensor Product $\otimes$, Number of Channels $C$ (for Gaunt tensor product).
  **procedure** LEARNABLETP($\mathbf{x}_1, \mathbf{x}_2$)
    **if** $\otimes = \otimes_{\text{CG}}$ **then**
      **return** LINEAR($\mathbf{x}_1 \otimes_{\text{CG}} \mathbf{x}_2$)
    **if** $\otimes = \otimes_{\text{GTP}}$ **then**
      **for** $i = 1, 2, \ldots, C$ **do**
        $\mathbf{x}_1^{(i)} \leftarrow$ LINEAR$_1^{(i)}(\mathbf{x}_1)$
        $\mathbf{x}_2^{(i)} \leftarrow$ LINEAR$_2^{(i)}(\mathbf{x}_2)$
        $\mathbf{x}_o^{(i)} \leftarrow$ LINEAR$_o^{(i)}(\mathbf{x}_1^{(i)} \otimes_{\text{GTP}} \mathbf{x}_2^{(i)})$
      **return** CONCATENATE($\{\mathbf{x}_o^{(i)} \mid i \in \{1, 2, \ldots, C\}\}$)
  **return** LearnableTP

---
**Algorithm 2** Operation of our Message Passing Neural Network
---

**Require:** Graph $G$, Message Passing Iterations $T$, Cutoff $d_{\max}$, Spherical Harmonic Degree $\ell$, Tensor Product $\otimes$
  Compute neighbor lists for each node in $G$:

$$(u, v) \in E \iff \|\mathbf{r}_u - \mathbf{r}_v\| \leq d_{\max}$$

  Create LEARNABLETENSORPRODUCT from $\otimes$.
  **for** $v \in V$ **do**:
    $h_v^{(0)} \leftarrow [1]$
  **for** $t = 1, 2, \ldots, T$ **do**:
    **for** $v \in V$ **do**:
      $h_v^{(t)} \leftarrow \frac{1}{|\mathcal{N}(v)|} \sum_{u \in \mathcal{N}(v)} \text{MLP}(\|\mathbf{r}_u - \mathbf{r}_v\|) \times$ LEARNABLETENSORPRODUCT($h_u^{(t-1)}, Y_\ell(\mathbf{r}_u - \mathbf{r}_v)$)
      $h_v^{(t)} \leftarrow$ GATE($h_v^{(t)}$)
      $h_v^{(t)} \leftarrow$ CONCATENATE($[h_v^{(t-1)}, h_v^{(t)}]$)
      $h_v^{(t)} \leftarrow$ LINEAR($h_v^{(t)}$)
  **return** $\{h_v^{(T)}\}_{v \in V}$

