# OpenReview forum: "Asymptotically Fast Clebsch-Gordan Tensor Products with Vector Spherical Harmonics"
_ICML.cc/2026/Conference — ICML 2026 regular_

### Official Review · Reviewer_KCwY · 2026-02-22

**Soundness:** 4
**Presentation:** 3
**Significance:** 4
**Originality:** 4
**Overall Recommendation:** 5
**Confidence:** 5

**Summary:**

This paper proposes a novel theoretical method (VSTP) that reduces the complexity of the Clebsch–Gordan tensor product from $\mathcal O(L^6)$ to $\mathcal O(L^4 \log^2 L)$ without sacrificing expressive power. The authors construct a complete and asymptotically superior computational framework by introducing tensor spherical harmonics and reinterpreting the Gaunt tensor product from the perspective of group Fourier transform. The theoretical derivation is rigorous and addresses the lack of antisymmetric interactions in existing methods.

**Compliance With Llm Reviewing Policy:**

Affirmed.

**Final Justification:**

See rebuttal acknowledgement.

**Key Questions For Authors:**

See above.

**Limitations:**

Yes.

**Strengths And Weaknesses:**

The mathematical derivations and theoretical analysis in this paper are natural, elegant, and easy to understand. I tend to accept this paper, and therefore I will point out more of the shortcomings of the current version, but these do not diminish the paper's contribution.

Although this paper is theoretical, it involves the analysis of algorithm complexity. I believe simple numerical experiments (even on a toy dataset and $L\leq 11$) are necessary to allow readers to intuitively understand the complexity curve and the constant factor. I think it would be appropriate for the authors to provide a comparison of the complexity growth curves of their algorithm with common operators such as the complete $\mathcal{O}(L^6)$ tensor product, the eSCN-optimized $\mathcal{O}(L^3)$ tensor product (including grid sizes for different SFFTs), and the $\mathcal{O}(L^2)$ inner product.
In addition, some supplementary references may improve the quality of the article:
- Spherical-scalarization models: SO3KRATES [A], HEGNN [B], GotenNet [C], which only use the tensor product path of inner product with $\mathcal{O}(L^2)$ complexity, but still achieve good results.
- Expressive power analysis: tensor product ensures model completeness [D], constructs different function bases [E], and maintains the model mapping space [F].

Finally, some minor suggestions: the coefficients appearing in the main text are quite lengthy (which is admittedly common in this line of work) and are not central to the main narrative. Combined with ICML’s two-column format, this makes parts of the paper visually dense and difficult to read. During the camera-ready stage, the authors might consider replacing some detailed coefficient expressions with the proportionality symbol $\propto$ in the main text and moving the full formulations to the appendix, which would substantially improve readability.

In addition, it would be helpful to include an algorithm block summarizing the proposed method. This would make the paper more accessible to readers who wish to implement the approach but may not have a strong theoretical background.

Overall, this is a nice article. I would be very happy to further improve my score if the aforementioned shortcomings are addressed.

> Reference
- [A] A Euclidean transformer for fast and stable machine learned force fields
- [B] Are High-Degree Representations Really Unnecessary in Equivarinat Graph Neural Networks?
- [C] GotenNet: Rethinking Efficient 3D Equivariant Graph Neural Networks
- [D] On the Universality of Rotation Equivariant Point Cloud Networks
- [E] Universally Invariant Learning in Equivariant GNNs
- [F] Reducing Symmetry Increase in Equivariant Neural Networks

---

> ### Author Rebuttal · Authors · 2026-03-31
>
> We thank the reviewer for their careful reading of our work and detailed feedback. We are glad the reviewer feels positively about our work and finds the derivations natural, elegant, and easy to understand.
>
> Regarding experiments, we implemented our method using the e3nn-jax framework [here](https://anonymous.4open.science/r/vector-spherical-harmonics-423F/README.md) with detailed pseudocode also provided. Our [implementation](https://anonymous.4open.science/r/vector-spherical-harmonics-423F/freedom/tensor_products/vector_spherical_harmonics.py) is quite simple and we also provide an easy to use API as seen in this [example notebook for VSTP](https://anonymous.4open.science/r/vector-spherical-harmonics-423F/vstp.ipynb).
>
> Then, we added a simple test on predicting the class of chiral 3D Tetris pieces using an equivariant neural network but with different 'tensor product' operations. This task was shown to be impossible for GTP in [1] but our VSTP was able to achieve perfect accuracy, similar to CGTP: https://anonymous.4open.science/r/vector-spherical-harmonics-423F/plots/tetris-results.png.
>
> Regarding complexity growth curves, we performed microbenchmarks similar to those in [1] comparing VSTP against GTP and CGTP. Note that these use SH transforms from `e3nn-jax` which are not the asymptotically fast versions. We are not aware of sub $O(L^3)$ SH transforms written in Python. However, we are able to confirm that VSTP is essentially a constant factor slower than GTP as expected from theory: https://anonymous.4open.science/r/vector-spherical-harmonics-423F/benchmarking/plots/walltime_gpu_MIMO_10000_icml.png.
>
> From our understanding, the eSCN trick only works if one of the irreps comes from the output of a SH; for a generic irrep feature we do not obtain the necessary sparsity. It is not clear how to adapt this for the scenario involving generic irrep features such as what happens in self interactions of MACE [2]. Lastly, we could add the inner product as well, but it is unclear to us how relevant this is as we expect it to be used quite differently in models unlike CGTP, GTP, and VSTP which can almost be dropin replacements for each other.
>
> We appreciate the suggestion of discussing scalarization models [A,B,C]. Indeed, the cost of CGTP has inspired many alternative approaches. We also appreciate the suggestion of articles looking at expressivity of equivariant models [D,E,F]. They can help motivate why we still want to have tensor products.
>
> For the coefficients, we will definitely consider this suggestion. However, for Theorem 4.3 in particular, we would like to keep the full expression as it could potentially be useful outside the equivariant ML community.
>
> Lastly, the suggestion of pseudocode is a good one. We will make sure to add the VSTP description from our repo above, following the structure outlined in section 4.3 (lines 306-319 col 1).
>
> We once again thank the reviewer for their suggestions. Please let us know if you have any other concerns or suggestions.
>
>
> ### References
> [1] Xie, Y., Daigavane, A., Kotak, M., & Smidt, T. The Price of Freedom: Exploring Expressivity and Runtime Tradeoffs in Equivariant Tensor Products. In Forty-second International Conference on Machine Learning.
>
> [2] Batatia, I., Kovacs, D. P., Simm, G., Ortner, C., & Csányi, G. (2022). MACE: Higher order equivariant message passing neural networks for fast and accurate force fields. Advances in neural information processing systems, 35, 11423-11436.

---

> > ### Author Rebuttal · Reviewer_KCwY · 2026-04-01
> >
> > I read all the discussions and replies, and I found the author's explanations to be very clear (especially regarding eSCN optimization and the inner product method, although I recommend keeping some simple textual descriptions), which has improved my rating.

---

> > > ### Author Response · Authors · 2026-04-02
> > >
> > > We are glad you found our explanations very clear and we will keep some simple textual descriptions in the final version. We thank you again for your feedback and for updating your rating.

---

### Official Review · Reviewer_KfMy · 2026-03-13

**Soundness:** 2
**Presentation:** 2
**Significance:** 2
**Originality:** 2
**Overall Recommendation:** 5
**Confidence:** 1

**Summary:**

This work focuses on processing interactions of group representations. Specifically, group representations can be decomposed into irreducible representations (irreps). It is useful to express them in a basis where each representation is a direct sum of irreps. Interactions between different irreps can be computed using tensor products, but this destroys the direct-sum structure. To recover the desired representation, a change of basis is required. This is done using Clebsch-Gordan (CG) coefficients. The entire process is called the Clebsch-Gordan tensor product (CGTP). Although a theoretical lower bound of the complexity of this method is O(L^4), typical implementations report a complexity O(L^6).

This paper proposes an algorithm that truly provides asymptotic gains, as it can be used to simulate full CGTP in O(L^4 log^2 L) time, without missing interactions and losing in expressivity as other alternatives. This is achieved by generalizing fast Fourier based convolution in a way that naturally leads to a modification of the Gaunt tensor product (GTP). GTP has been proposed in the past as a fast but incomplete alternative of CGTP, since interactions can be missed. Then the authors show that the suboptimality of GTP rising from antisymmetry, can be resolved by switching to irrep valued signals and working with tensor harmonics. Through that transition, they propose a generalized Gaunt formula for the tensor harmonics. Based on this result, they show that vector spherical harmonics and vector valued signals suffice from recovering efficiently the missing interactions of the GTP.

**Compliance With Llm Reviewing Policy:**

Affirmed.

**Final Justification:**

After considering the authors’ rebuttal and the reviewer discussion, my assessment has improved. The authors responded satisfactorily to my comments and addressed most of the questions raised during the discussion. While my own confidence in this area is limited, I also took into account the detailed feedback from other reviewers. There is general agreement that the theoretical contributions are interesting and significant. In particular, reviewer KCwY, who has high confidence, provided a strong positive assessment, which I find convincing. Although empirical validation is currently missing and I would strongly encourage the authors to include it in a final version, I am persuaded that the theoretical contributions are substantial. Based on this, I increase my score to 5.

**Key Questions For Authors:**

Question 1: The authors mention in the paper that: “there are asymptotically fast SH transform algorithms which allow the Gaunt tensor product (GTP) (Luo et al., 2024) to achieve additional asymptotic runtime gains”. However, it is not clear whether these algorithms also rely on FFT convolutions. If they do, the positioning of section 3 should be different. If not, what are the advantages of the FFT based approach and how do the different approaches on GTP compare in terms of complexity? In others words, why is Section 3 important if there are other ways to compute GTPs?

Question 2:  The authors mention that the corresponding Nyquist sampling theorem requires the number of points to scale as L^3, which I understand induces a complexity of O(L^3) by itself. However, this scaling does not seem to be reflected in the decomposition described in Section 6.2. I would appreciate clarification on this point.

**Limitations:**

Yes

**Strengths And Weaknesses:**

Most parts of this work are clearly written and well structured. However, the narrative is quite technical, which makes it difficult to follow the intuition. For example, the connection between the previously proposed Gaunt tensor product and group Fourier transforms is presented clearly. Still, it is not obvious why this connection is useful for the rest of the paper, even though it spans roughly a quarter of the text.

The submission appears to be technically sound, at least to the extent that my background allows me to evaluate it. The authors seem careful and honest in assessing both the strengths and weaknesses of their work. They also properly position their work in the context of prior and concurrent literature. However, they do not clearly discuss how it differs in some aspects (see my questions).

This paper addresses an important problem, which could potentially advance understanding, capabilities, or practice in applied group theory and, by extension, machine learning. It could influence future research or applications, as it aims to accelerate the computation of a fundamental concept.

Given the validity of the presented claims, this work provides new insights and deepens understanding by introducing new theory. Some parts of the contributions are not clearly distinguished from closely related literature. However, the novelty of the work is still well justified.

---

> ### Author Rebuttal · Authors · 2026-03-31
>
> We thank the reviewer for their efforts in understanding our work and their helpful feedback. We are glad the reviewer finds the paper clearly written, well structured, and technically sound.
>
> ### Questions
>
> > The authors mention in the paper that: “there are asymptotically fast SH transform algorithms which allow the Gaunt tensor product (GTP) (Luo et al., 2024) to achieve additional asymptotic runtime gains”. However, it is not clear whether these algorithms also rely on FFT convolutions. If they do, the positioning of section 3 should be different. If not, what are the advantages of the FFT based approach and how do the different approaches on GTP compare in terms of complexity? In others words, why is Section 3 important if there are other ways to compute GTPs?
>
> We believe there may be confusion between standard Fourier transforms (FT) and general group Fourier transforms. The bottleneck of GTP occurs in the SH transform step and the original proposal has some extra standard FT steps. Various SH transform algorithms also use standard FFT as a subroutine but are bottlenecked by the associated Legendre transform (ALT) step. Semi-naive ALT gives the final $O(L^3)$ runtime and is more commonly used but there are recursion relations allowing $O(L^2\log^2 L)$ runtime [1].
>
> The point of section 3 is to **explain why GTP was a natural thing** to try from the perspective of group FT. We are motivated by the use of standard FT ($SO(2)$ version of general group FT) to accelerate convolutions (which is a $SO(2)$ analog of $SO(3)$ tensor product). The natural question is then whether we can use $SO(3)$ group FT (section 3.1) in a similar way and if there is a relevant convolution theorem (section 3.2). We run into some issues (one of which is your Q2) which motivates using a quotient ($SO(3)/SO(2)\cong S^2$ in this case) (section 3.3) instead. This naturally leads to GTP. Importantly, this entire process can naturally be done for other groups (replace $SO(3)$ with arbitrary $G$ and replace $SO(2)$ with some subgroup $H<G$), giving a recipe for Gaunt-like operations on other groups. See our response to reviewer DNyn for a detailed sketch of how the process changes for other groups.
>
> In contrast, the original GTP paper was motivated by Wigner-Eckart and existence of Gaunt formula. It was **not originally obvious how this would generalize** to other groups nor why one might expect it to give a fast algorithm.
>
> > The authors mention that the corresponding Nyquist sampling theorem requires the number of points to scale as L^3, which I understand induces a complexity of O(L^3) by itself. However, this scaling does not seem to be reflected in the decomposition described in Section 6.2. I would appreciate clarification on this point.
>
> This is a good question. The $O(L^3)$ comes from the fact that $SO(3)$ is a 3-dimensional manifold. This is the motivation for section 3.3 where the known fact that taking a quotient with a subgroup gives the sphere $SO(3)/SO(2)\cong S^2$, giving a 2-dimensional manifold. On the sphere, the number of sample points required is now $O(L^2)$ so it is no longer a bottleneck.
>
>
> We hope this response helps clarify your questions. Please let us know if you have any other concerns.
>
> ### References
>
> [1] Healy Jr, D. M., Rockmore, D. N., Kostelec, P. J., & Moore, S. (2003). FFTs for the 2-sphere-improvements and variations. Journal of Fourier analysis and applications, 9(4), 341-385.

---

> > ### Author Rebuttal · Reviewer_KfMy · 2026-04-02
> >
> > The authors have satisfactorily addressed all my questions. Thank you.

---

### Official Review · Reviewer_DNyn · 2026-03-15

**Soundness:** 3
**Presentation:** 3
**Significance:** 2
**Originality:** 3
**Overall Recommendation:** 4
**Confidence:** 3

**Summary:**

The authors present a theoretically novel way to accelerate CGTP. They employ tensor spherical harmonics to alleviate the expressivity and antisymmetry problems associated with Gaunt tensor products. This approach leads them to a generalized Gaunt formula. They show that by using only vector signals, rather than higher-degree tensors, all valid $(j_1, j_2, j_3)$ paths for CGTP are interactable, except for the trivial scalar (zero) case. The proposed tensor products, vector signal tensor products (VSTP) interprets irreps as vector spherical signals, performs pointwise cross products, and projects back to harmonic coefficients. Finally, they establish a runtime reduction from $\mathcal{O}(L^6)$ to $\mathcal{O}(L^4 \log^2 L)$ for simulating full CGTP using their proposed VSTP.

**Compliance With Llm Reviewing Policy:**

Affirmed.

**Final Justification:**

I believe the overall theoretical contribution is interesting and significant. However, the practical relevance of the proposed method is conditioned on existence of fast SH transforms which are currently unavailable as authors suggest. As such, I believe my initial rating of weak accept is appropriate and I am maintaining that.

**Key Questions For Authors:**

1. In L192 (right column): Do you mean arbitrary Lie groups, or arbitrary compact (and possibly connected) Lie groups?
2. In L221 (left column): What does the notation $Y_l^{m,*}$ denote?
3. Did you attempt to implement the proposed method?  If so, what specific issues were encountered?

**Limitations:**

Yes

**Strengths And Weaknesses:**

### Strengths
* The employment of tensor spherical harmonics and the derivation of the generalized Gaunt formula are original contributions.
* Demonstrating that only vector signals are needed, rather than higher-order tensors, significantly reduces the additional complexity introduced by Tensor Spherical Harmonics (TSH). The selection rules of Vector Signal Tensor Products (VSTP) neatly establish this.
* The CGTP simulation using VSTPs, which improves the theoretical complexity from $\mathcal{O}(L^6)$ to $\mathcal{O}(L^4 \log^2 L)$, is a significant reduction.

### Weaknesses
* **Significance:** A major weakness is the lack of empirical evidence to support the theoretical claims of an asymptotically fast CGTP. Furthermore, the authors do not clearly explain the absence of this empirical validation.
* The proposed asymptotic improvement may not translate to practical gains in current equivariant networks where
$L$ is typically small.
*  The claimed contribution of an "explicit connection to group Fourier Transforms, allowing generalization to other groups" is not highly substantial, nor is it thoroughly supported by the rest of the paper. The connection to GFT already exists in the literature, and the generalization to other groups is not explicitly demonstrated.


### Comments and Suggestions
* **L142 (right column):** Please clearly define what $d_\pi$ is.
* **L151 (right column):** In the RHS equation, shouldn't $D_{ij}^{(\pi)}(g)$ actually be $D_{ij}^{(\pi)*}(g)$?
* **L227 (left column):** Typo in the LHS; $m_2$ is missing.
* **L281 (right column):** The term $C_{l_1, 0, l_2, 0}^{l, 0}$ should likely be $C_{l_1, 0, l_2, 0}^{l_3, 0}$.

---

> ### Author Rebuttal · Authors · 2026-03-31
>
> We thank the reviewer for their careful reading and helpful feedback. We are glad the reviewer find our reduction significant and our method neat.
>
> ### Weaknesses
> > A major weakness is the lack of empirical evidence to support the theoretical claims of an asymptotically fast CGTP. Furthermore, the authors do not clearly explain the absence of this empirical validation.
>
> We explain in Section 7 that current model sizes are too small to benefit from the asymptotic gains. Further, the asymptotic results rely on a fast SH transform, but we are not aware of any sub $O(L^3)$ spherical harmonic transform libraries implemented in Python. (The implementation of fast spherical harmonic transforms is particularly tricky due to the need for a specific data structure for convolution masks + repeated FFT calls for polynomial multiplication, as described in [Driscoll and Healy](https://digitalcommons.dartmouth.edu/cgi/viewcontent.cgi?article=1205&context=cs_tr)). However, we did implement a version of VSTP using the SH transform present in the `e3nn-jax` library and performed microbenchmarking similar to that described in [1]. While this is unable to show the full asymptotic gains, we were able to show that VSTP is a constant factor slower than GTP as expected from the theory (https://anonymous.4open.science/r/vector-spherical-harmonics-423F/benchmarking/plots/walltime_gpu_MIMO_10000_icml.png).
>
> > The claimed contribution of an "explicit connection to group Fourier Transforms, allowing generalization to other groups" is not highly substantial, nor is it thoroughly supported by the rest of the paper. The connection to GFT already exists in the literature, and the generalization to other groups is not explicitly demonstrated.
>
> This is a good question. Perhaps it is more accurate to state we use the connection to understand tensor products. As far as we are aware, prior literature mostly shows equivalence between group convolution and irrep based linear layer constructions which we mention (lines 182-195 column 1). We are not aware of equivariant literature that explicitly explains how tensor products show up from pointwise multiplication in real space.
>
> To the best of our knowledge, GTP is the first to use pointwise multiplication but they are motivated by Wigner-Eckart theorem and existing knowledge of Gaunt coefficients. The purpose of section 3 is to explain why GTP was the natural thing to try based off of trying to generalize the idea behind FFT convolutions. See our response to Q1 of reviewer KfMy for a high level overview of this motivation.
>
> We sketch in more detail how section 3 changes for other compact semisimple Lie groups. First, the **structure of equation (2) remains unchanged** and we replace $D$ with the irrep transformation matrices of the other group and $C$ with the Clebsch-Gordan coefficients (for chosen irrep basis) with CG coefficients of the other group. This is because we can derive (2) by viewing the left side $D_{m_1,n_1}^{(\ell_1)}D_{m_2,n_2}^{(\ell_2)}$ as an object which has tensor product reps for left and right group action which we can decompose into irreps using CG coefficients on the pair of left ($m_1,m_2$) and right ($n_1,n_2$) indices respectively. For simplifying $D^{(\ell)}_{m,n}$ to form a basis for a quotient manifold (line 222 col 1), this can likely be done if we chose the subgroup to be the maximal torus (its Lie algebra would be a Cartan subalgebra $\mathfrak{h}$). If the irrep basis is compatible with weight space decomposition, we only keep indices which have weight $0$ (in $SO(3)$ case this is setting $n=0$). We can provide a more in depth discussion of this generalization strategy in the appendix.
>
> ### Comments and Suggestions
> * $d_\pi$ is dimension of the irrep
> * Yes, thanks for catching this!
> * Yes, thanks for catching this!
> * Yes, thanks for catching this!
>
> ### Key Questions
> 1. Good question. For us, it is clear how to proceed for compact semi-simple Lie groups and we will modify the statement accordingly. It may be possible to do for other Lie groups as well.
> 2. We mean complex conjugation by $*$.
> 3. Yes, we implemented our method using the e3nn-jax framework [here](https://anonymous.4open.science/r/vector-spherical-harmonics-423F/README.md) with detailed pseudocode also provided. Our [implementation](https://anonymous.4open.science/r/vector-spherical-harmonics-423F/freedom/tensor_products/vector_spherical_harmonics.py) is quite simple, see [example notebook for VSTP](https://anonymous.4open.science/r/vector-spherical-harmonics-423F/vstp.ipynb). See also our response to reviewer KCwY where we perform some initial tests and benchmarks with VSTP.
>
> We hope this response helps address your concerns. Please let us know if you have any other questions.
>
> ### References
> [1] Xie, Y., Daigavane, A., Kotak, M., & Smidt, T. The Price of Freedom: Exploring Expressivity and Runtime Tradeoffs in Equivariant Tensor Products. In Forty-second International Conference on Machine Learning.

---

> > ### Author Rebuttal · Reviewer_DNyn · 2026-04-02
> >
> > Thank you for the rebuttal. I believe the overall theoretical contribution is interesting and significant. However, the practical relevance of the proposed VSTP is conditioned on existence of fast SH transform which are currently unavailable as you suggest. As such, I believe my initial rating is fair and I am maintaining that. I believe the reasons to accept this paper marginally outweigh reasons to reject it. Hence, I am going with weak accept.

---

> > > ### Author Response · Authors · 2026-04-02
> > >
> > > We once again thank the reviewer for their feedback. We agree that the asymptotic gains from using VSTP to simulate CGTP (depending on fast SH) are currently impractical but are theoretically interesting.
> > >
> > > However, VSTP on its own is still $O(L^3)$ even without the fast SH algorithms, the same as GTP which has proven to be useful. We also demonstrated that it is a constant factor slower than GTP but recovers the antisymmetric paths. Hence, VSTP could be practical as a CGTP alternative for tasks where such antisymmetric paths are important. We mentioned this point in Section 7 (lines 416-426 col 1).

---

### Decision · Program_Chairs · 2026-04-30

**Decision:**

Accept (regular)

**Comment:**

The paper proposes a theoretically justified algorithm to reduce complexity of Clebsch-Gordan transform from $\mathcal{L}^6$ to $\mathcal{L}^4 \log^2 (L)$. This is an interesting result, and two reviewers recommend acceptance. The concern of the third (negative) reviewer is that there are no numerical experiments with the author answer that 'the model sizes are too small for this to be faster'. Both are fair points, but I think a fast algorithm is machine learning without a practical application somehow reduces the value of the contribution.